# EFFECTIVE PASSIVE MEMBERSHIP INFERENCE AT­TACKS IN FEDERATED LEARNING AGAINST OVERPA­RAMETERIZED MODELS

**Jiacheng Li, Ninghui Li & Bruno Ribeiro**
Department of Computer Science
Purdue University
West Lafayette, IN 47903, USA
`li2829@purdue.edu,{ninghui,ribeiro}@cs.purdue.edu`

## ABSTRACT

This work considers the challenge of performing membership inference attacks in a federated learning setting —for image classification— where an adversary can only observe the communication between the central node and a single client (a passive white-box attack). Passive attacks are one of the hardest-to-detect attacks, since they can be performed without modifying how the behavior of the central server or its clients, and assumes *no access to private data instances*. The key in­sight of our method is empirically observing that, near parameters that generalize well in test, the gradient of large overparameterized neural network models statis­tically behave like high-dimensional independent isotropic random vectors. Us­ing this insight, we devise two attacks that are often little impacted by existing and proposed defenses. Finally, we validated the hypothesis that our attack depends on the overparametrization by showing that increasing the level of overparametriza­tion (without changing the neural network architecture) positively correlates with our attack effectiveness.

## 1 INTRODUCTION

Our work considers the challenge of performing membership-inference (MI) attacks —for image classification— in a federated learning setting, where an adversary can only observe the commu­nication between the central node and a single client (a passive white-box attack). We will also consider, in passing, other attack modalities (e.g., active white box attacks) but our focus will be on passive attacks. Passive attacks are one of the hardest-to-detect attacks, since they can be performed without modifying the behavior of the central server or its clients, and *assumes **no access** to private data instances*. Our results consider multiple applications, but pay special attention to medical im­age diagnostics, which is one of the most compelling and most sensitive applications of federated learning.

Federated learning is designed to train machine-learning models based on private local datasets that are distributed across multiple clients while preventing data leakage, which is key to the develop­ment of machine learning models in medical imaging diagnostics (Sheller et al., 2020) and other industrial settings where each client (e.g., hospital, company) is unwilling (unable) to share data with other clients (e.g., other hospitals, companies) due to confidentiality laws or concerns, or in fear of leaking trade secrets. Federated learning differs from distributed data training in that the data may be heterogeneous (i.e., each client data is sampled from different training distributions) and client's data must remain private (Yang et al., 2019). From the attacker's perspective, the hardest membership attack setting is one where the client's data is sampled from the training distribution (the i.i.d. case), since in this scenario there is nothing special about the data distribution of any spe­cific client that an attacker could use as leverage. Our work focuses on the this i.i.d. case. As far as we know there are no known passive white-box membership inference attack specifically designed to work in a scenario without private data access in federated learning (the closest works (Nasr et al., 2019; Zari et al., 2021) are actually very different because they assume access to private data). Unfortunately, we show that for large deep learning models (overparameterized models), there is a

membership inference attack that leverages the foundation of how we train these large models. Our proposed attack works, for instance, on systems secured by additive homomorphic encryption (Aono et al., 2017), where the central server knows nothing about each client's gradients. The attack just needs access to the central model parameters (decoded by clients) at each communication round. Interestingly, our attack relies on statistical properties of overparameterized models (calibrated via conformal predictors) and will not work as well in small neural network models. We detail our attack next.

**Attack insight:** Large overparameterized models perform surprisingly well on test data even though they can overfit the training data, creating learning phenomena such as double descent (Nakkiran et al., 2021). In machine learning, this overfitting needs to be kept in check to avoid black-box membership attacks (Li et al., 2020; Yeom et al., 2018), but there are simple regularization solutions to address this challenge (e.g., (Li et al., 2020; Nasr et al., 2018)). Overparameterized models are hypothesized to perform well in test because the learning finds wide flat minima (in a robust definition of wide (e.g., (Neyshabur et al., 2018)), which is non-trivial (Dinh et al., 2017)), often attributed to gradient noise in gradient descent and overparameterization (Baldassi et al., 2016; Keskar et al., 2016; Martin & Mahoney, 2021; Neyshabur et al., 2014; Zhang et al., 2021).

Let's take a closer look at the optimization of overparameterized models with $d \gg 1$ parameters. In our setting, an overparameterized model has significantly more parameters than training instances. Through an extensive **empirical** evaluation (described in Section 4) shows that, at later gradient update rounds $t \gg 1$ of the optimization (in our experiments $t > 50$ if trained from scratch and $t \geq 2$ if fine-tuned) of medium to large neural networks —and at nearly any stage of the fine tuning of large pre-trained models— gradient **vectors** of different training instances are orthogonal in the same way distinct samples of *independent isotropic random vectors* are orthogonal (such as two high-dimensional Gaussian random vectors with zero mean and diagonal covariance matrix (isotropic)). More precisely, the vectors in $\mathcal{V} = \{\widetilde{\nabla}_{y_i, \boldsymbol{x}_i, t}\}_{i=1}^n$ become nearly orthogonal as $d \gg 1$, where $\widetilde{\nabla}_{y, \boldsymbol{x}, t} = \nabla_{\boldsymbol{W}^{(t)}} \mathcal{L}(y, \boldsymbol{x}; \boldsymbol{W}^{(t)}) / \|\nabla_{\boldsymbol{W}^{(t)}} \mathcal{L}(y, \boldsymbol{x}; \boldsymbol{W}^{(t)})\|_2$ with loss $\mathcal{L}$, model parameters $\boldsymbol{W}^{(t)} \in \mathbb{R}^d$, and training data $\mathcal{D}_{\text{train}} = \{(y_i, \boldsymbol{x}_i)\}_{i=1}^n$. That is, for large models ($d \gg 1$) and for large enough $t$, the normalized gradients are approximately orthogonal $\langle \widetilde{\nabla}_{y_i, \boldsymbol{x}_i, t}, \widetilde{\nabla}_{y_j, \boldsymbol{x}_j, t} \rangle \approx 0$, $i \neq j$ (see Figure 1(a-c)), where $\langle \cdot, \cdot \rangle$ is the inner product. But for small $d$, the normalized gradients no longer can be relied to be approximately orthogonal (as we will infer in Figure 3). These results match the property of independent $d$-dimensional isotropic random vectors, whose independent samples become increasingly orthogonal at a rate $\langle \widetilde{\nabla}_{i,t}, \widetilde{\nabla}_{i,t} \rangle \in O(1/\sqrt{d})$ (Vershynin, 2018, Lemma 3.2.4).

Hence, given a set of (largely unknown) orthogonal vectors $\mathcal{O}$ and a private subset $\mathcal{O}' \subset \mathcal{O}$, by the distributive property of inner products we have $\forall \boldsymbol{u} \in \mathcal{O}, \langle \boldsymbol{s}_{\mathcal{O}'}, \boldsymbol{u} \rangle > 0$ if and only if $\boldsymbol{u} \in \mathcal{O}'$, where $\boldsymbol{s}_{\mathcal{O}'} = \sum_{\boldsymbol{v} \in \mathcal{O}'} \boldsymbol{v}$. This means that for a large model trained with a gradient update $\boldsymbol{W}^{(t)} = \boldsymbol{W}^{(t-1)} - \eta \Gamma^{(t)}$ performed at the central server, where $\Gamma^{(t)}$ is the sum (or average) of all client gradients at communication round $t$, as long as we have access to the previous model $\boldsymbol{W}^{(t-1)}$ and the new model $\boldsymbol{W}^{(t)}$, we obtain $\eta \Gamma^{(t)} = \boldsymbol{W}^{(t-1)} - \boldsymbol{W}^{(t)}$ from which we can test whether an instance $\boldsymbol{x}$ belongs to the private data by asking whether $\langle \eta \Gamma^{(t)}, \widetilde{\nabla}_{y, \boldsymbol{x}, t-1} \rangle > 0$. Later in the paper we show a different type of attack (using subspaces and the $L_2$ norm) that does not need inner products.

To summarize our findings: *The almost orthogonality of gradients of independent training examples in overparameterized models means gradient sums cannot hide a specific gradient.* This attack is difficult to defend since it seeks to attack the foundation of how overparameterized models learn to generalize over the test data. In our experiments we have tried a host of defenses with only moderate success: (a) Adding isotropic Gaussian noise $\epsilon$ (the DP-SGD defense (Abadi et al., 2016)) is innocuous since the noise is also orthogonal to the gradients (i.e., $\langle \epsilon, \widetilde{\nabla}_{y_i, \boldsymbol{x}_i, t-1} \rangle \approx 0$ for all $(y_i, \boldsymbol{x}_i) \in \mathcal{D}_{\text{train}}$). (b) If the noise is biased (e.g., non-isotropic noise), it will bias the gradients and give poor model generalization. (c) Quantizing all client gradients (e.g., signSGD (Bernstein et al., 2018)) makes the attack less effective but it is not enough to prevent the attack altogether (the quantized gradients are also nearly isotropic). (d) Borrowing ideas from meta-learning (MALM (Finn et al., 2017) more exactly), we are able to reduce the effectiveness of the attack at the expense of poorer generalization error. (e) Changing how the central model performs gradient updates relies on proving the attacker will never be able to find a vector proportional to $\epsilon + e(G)$ from the model updates, where $\epsilon$ is approximately isotropic noise, $G$ is the gradient sum of all clients, and $e(G)$ is a

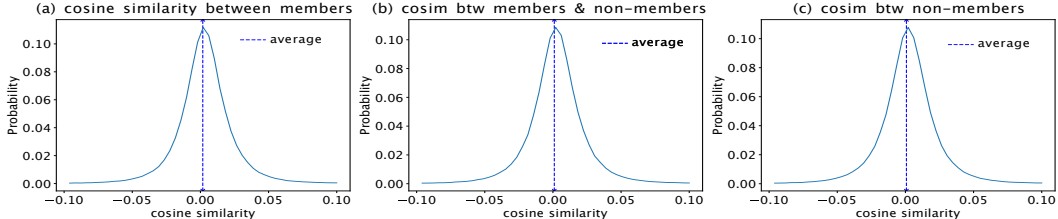

Figure 1: Representative example of our findings (at epoch 250, AlexNet, CIFAR-100 dataset): (a) Distribution of cosine similarity (cosim) between gradients of two distinct member instances. (b) Distribution of cosine similarity between gradients of one member instance and one non-member instance. (c) Distribution of cosine similarity between gradients of two distinct non-member instances.

vector in the direction of $G$. Having model updates that lead to good generalization but yet do not follow the direction of the gradient $G$ is a difficult feat, one which we were unable to achieve in this work. More research is needed in order to defend against this dangerous attack.

**Contributions:** The primary goal of our paper is not to exhaust all possible defenses against this attack (they may exist). Rather, the primary goal of our paper is to make the following contributions:

1. We characterize and provide a systematic analysis of gradient orthogonality of data instances in overparameterized neural networks, which has been analyzed in bits and pieces (Mirzadeh et al., 2022) with respect to multiple tasks and catastrophic forgetting, but never in a systematic way for a single task.

2. Then, inspired by the gradient orthogonality phenomenon we characterized, we propose two highly effective passive membership inference attacks against federated learning that require only small (non-member) validation datasets and *do not require* private (member) data (unlike most existing attacks). We show that these attacks can be quite effective against diverse tasks (including medical imaging and structured data tasks) and against existing defenses (such as gradient quantization).

## 2 PROPOSED PASSIVE MI ATTACKS FOR OVERPARAMETERIZED NEURAL NETWORKS

### 2.1 FEDERATED LEARNING

This work focuses on membership-inference (MI) attacks classifiers trained by federated learning. Each data instance is a pair $(\boldsymbol{x}, y)$, where $\boldsymbol{x} \in \mathbb{X}$ is the feature and $y \in \mathbb{Y}$ is the label from a set of discrete labels $\mathbb{Y}$. The training dataset $\mathcal{D}_{\text{train}}$ used to train a a neural-network softmax classifier $F_\theta$. On an input $x$, the classifier's output $F_\theta(x)$ is a probability distribution over $\mathbb{Y}$. Federated learning (Konečný et al., 2016; McMahan et al., 2017) was designed to accomplish the goal of training a central model using private data distributed on multiple devices. The total training dataset is distributed into $C$ clients, where each client $c \in \{1, \ldots, C\}$ owns a *private* dataset $\mathcal{D}_c$ which contains $N_c$ training instances. We assume we do not have access to any examples in $\mathcal{D}_c$. Our work considers the **FedratedAveraging** (McMahan et al., 2017) protocol, where the $t+1$-st communication round of training starts with the central server distributing the current central model with parameter $\boldsymbol{W}_S^{(t)}$ to the clients. Starting from the global parameters $\boldsymbol{W}_S^{(t)}$, client $c$ uses the local dataset to perform **one or more** gradient descent steps, arriving at a model $\boldsymbol{W}_c^{(t)}$, and then send to the server

$$\Delta \Gamma_c^{(t)} = \boldsymbol{W}_c^{(t)} - \boldsymbol{W}_S^{(t)}. \tag{1}$$

Finally, the server averages $\Delta \bar{\Gamma}^{(t)} = \sum_{c=1}^{C} \Delta \Gamma_c^{(t)} / C$ the received parameter updates, and apply the update $\boldsymbol{W}_S^{(t+1)} = \boldsymbol{W}_S^{(t)} + \Delta \bar{\Gamma}^{(t)}$.

### 2.2 EXISTING PASSIVE ATTACKS

Later in Section 3 and Appendix A.9 we provide an extensive literature review on membership-inference and federated learning attacks. Here, however, we emphasize attacks that can be per-

formed passively (by just passively monitoring the model updates). Blackbox attacks such as Yeom et al. (2018); Jayaraman et al. (2020) are, as far as we know, the only existing passive federated learning attacks that can work *without access to private information* and *without access to a large set of training examples from the training distribution that produced* $\mathcal{D}_{train}$ (from which one could build a shadow model). In typical federated learning settings that arise, for instance, when learning classifiers on medical images, the use of shadow models (Carlini et al., 2021; Shokri et al., 2017; Song et al., 2019; Sablayrolles et al., 2019; Long et al., 2020; Watson et al., 2021; Ye et al., 2021) is challenging, since it requires access to the training distribution and large training datasets (usually as large as the federated learning's $\mathbb{D}_{train}$). Similarly, attacks such as Zari et al. (2021) and Nasr et al. (2019) that require private information are also challenging since private data (such as medical images) tend to be in systems with strict privacy guarantees. We note in passing that our attack could also be adapted to use shadow models and private information, which we leave as future work.

### 2.3 PROPOSED PASSIVE MONITORING ATTACKS

Our key challenge is to effectively exploit gradient updates for MI attacks, without intervening with the model learning, without access to private data, and without access to examples of the training distribution. In the following discussion we target a particular communication round $t$ but we will omit $t$ in some equations for clarity. Since each client updates its local model through $K$ mini-batches, we must define the derivatives communicated by the clients using hidden $K$ half-step updates. We first note that client's $c \in \{1, \ldots, C\}$ update $\Gamma_c^{(t+1)}$ can be defined as the final iteration of $K$ half-step updates $\Delta\Gamma_c$ (performed at the client over the private data $\mathcal{D}_c$) for $k = 1, \ldots, K$,

$$\Delta\Gamma_c^{(t+k/K)} = \Delta\Gamma_c^{(t+(k-1)/K)} - \eta \sum_{(\boldsymbol{x},y)\in\mathcal{D}_c} \nabla_{\boldsymbol{W}}\mathcal{L}(y,\boldsymbol{x};\boldsymbol{W})\Big|_{\boldsymbol{W}=\boldsymbol{W}_S^{(t)}+\Delta\Gamma_c^{(t+(k-1)/K)}}, \quad (2)$$

where $\Delta\Gamma_c^{(t)} = \boldsymbol{0}$, $\mathcal{D}_c$ is the set of private training instances at client $c$, $\nabla_{\boldsymbol{W}}\mathcal{L}(y,\boldsymbol{x};\boldsymbol{W})$ denotes the gradient one gets when using $(y, \boldsymbol{x})$ to train the current client model $\boldsymbol{W}_S^{(t)} + \Delta\Gamma_c^{(t+(k-1)/K)}$ at half-step $k$, and $\eta$ is the learning rate (the reminder of the notation is standard calculus notation we assume the reader is familiar with). Beyond full-batch gradient descent (e.g., SGD, Adam, momentum) Equation (2) needs to be slightly modified, but the basic principles of gradient orthogonality we describe later remain roughly the same.

**Threat model:** We consider a passive eavesdropping attacker, who is able to observe updated model parameters from rounds of communication between server and clients. The attacker may observe either (a) the server $\to$ client (global) model update, (b) client $\to$ server model updates, or (c) both. In this work we focus on eavesdropping client $\to$ server model updates. Different from most existing works which assume access to private member data instances (Nasr et al., 2019; Zari et al., 2021), we assume that the attacker has access to a small validation dataset which contains only non-member data instances. Our work is motivated by the strict data handling protections of medical records, creating attack vectors that do not rely on having access to private medical data.

**Attacker's goal:** The goal of our attack is to predict the value of $M(\boldsymbol{x}, c) := \mathbb{1}\{(\cdot, \boldsymbol{x}) \in \mathcal{D}_c\}$, which is an indicator variable showing whether $\boldsymbol{x}$ is in the private dataset $\mathcal{D}_c$. For any non-trivial $\boldsymbol{W}$, the gradients $\nabla_{\boldsymbol{W}}\mathcal{L}(y,\boldsymbol{x};\boldsymbol{W})$ and $\nabla_{\boldsymbol{W}}\mathcal{L}(y,\boldsymbol{x}';\boldsymbol{W})$ in an overparameterized model (with non-trivial parameters $\boldsymbol{W}$) should be different for $\boldsymbol{x}' \neq \boldsymbol{x}$, and hence we can re-define $M(\boldsymbol{x}', c)$ as

$$M(\boldsymbol{x}', c) = \sum_{a\in\mathbb{Y}} \mathbb{1}\{\nabla_{\boldsymbol{W}}\mathcal{L}(a,\boldsymbol{x}';\boldsymbol{W}_S^{(t)}) \text{ is in the sum that defines } \Delta\Gamma_c^{(t+1)} \text{ (or } \Delta\bar{\Gamma}^{(t+1)})\}, \quad (3)$$

where we sum over $\mathbb{Y}$ since we assume we do not know the label of $\mathbf{x}'$ in the private data.

**Gradient orthogonality:** Answering Equation (3) requires us to understand how, for $a \in \mathbb{Y}$, the gradient vectors $\nabla_{\boldsymbol{W}}\mathcal{L}(a,\boldsymbol{x};\boldsymbol{W})$ and $\nabla_{\boldsymbol{W}}\mathcal{L}(a,\boldsymbol{x}';\boldsymbol{W})$ from different data instances $\boldsymbol{x} \in \mathbb{X}$ and $\boldsymbol{x}' \in \mathbb{X}$ relate to each other. In this investigation, we empirically found that —after enough training epochs— the overparameterized model gradients of different instances are nearly orthogonal, just like high-dimensional independent isotropic random vectors. Moreover, their cosine similarity is Gaussian. Figure 1 plots the distribution of pair-wise cosine similarity of gradients from different instances in one of our experiments, which is representative example of the remaining experiments described later.

We believe this phenomenon is due to the fact that at later stages in training, the gradient of each instance is a sparse, independent sample of a high dimensional distribution, and that, while overall

the gradients of different instances provide a general direction for descent to minimize the loss, there is a strong randomness component in these gradients, which is conjectured as one of the key reasons for the ability of gradient descent to generalize well in overparameterized neural networks (Dinh et al., 2017; Sagun et al., 2017; Zhang et al., 2021).

**Cosine attack:** From Equation (2) and assuming the orthogonality of gradients of different training instances, we can formally define Equation (3) based on the angles between the two vectors as follows (via cosine similarity, which also works if we replace $\Delta\Gamma_c^{(t+1)}$ by $\Delta\bar{\Gamma}^{(t+1)}$)

$$M(\boldsymbol{x}', c) = \sum_{a \in \mathbb{Y}} \mathbb{1}\{\text{cosim}(\nabla_{\boldsymbol{W}}\mathcal{L}(a, \boldsymbol{x}'; \boldsymbol{W}_S^{(t)}), \Delta\Gamma_c^{(t+1)}) \geq \gamma\}, \tag{4}$$

where

$$\text{cosim}(\nabla_{\boldsymbol{W}}\mathcal{L}(a, \boldsymbol{x}'; \boldsymbol{W}_S^{(t)}), \Delta\Gamma_c^{(t+1)}) = \frac{\langle \nabla_{\boldsymbol{W}}\mathcal{L}(a, \boldsymbol{x}'; \boldsymbol{W}_S^{(t)}), \Delta\Gamma_c^{(t+1)} \rangle}{(\|\nabla_{\boldsymbol{W}}\mathcal{L}(a, \boldsymbol{x}'; \boldsymbol{W}_S^{(t)})\|_2 \|\Delta\Gamma_c^{(t+1)}\|_2)},$$

and $\langle \cdot, \cdot \rangle$ again is the inner product, threshold $\gamma = 1$ if gradients of different instances are exactly orthogonal, but can be set $\gamma > 1$ higher if gradients are only nearly orthogonal. We will describe a procedure to set hyperparameter $\gamma$ *without* the need to access private instances in $\mathcal{D}_c$.

Figure 2(a) shows the difference between the distributions of the cosine similarity of instances from the training data $(\cdot, \boldsymbol{x}') \in \mathcal{D}_c$ (members) and from non-members $(\cdot, \boldsymbol{x}') \notin \mathcal{D}_c$. Note that both distributions look Gaussian but with different averages. Non-members have average nearly zero while members have average close to 0.01. Later we show that this small difference is enough to design an effective attack using multiple communication rounds.

**Gradient-diff (subspace) attack:** Now we propose another attack based on orthogonality. If $S = a + b + c$ is a sum of orthogonal vectors, then $\|S\|_2^2 - \|S - a\|_2^2 = \|a\|_2^2$, otherwise for another vector $f$ orthogonal to $a, b, c$ we have $\|S\|_2^2 - \|S - f\|_2^2 = -\|f\|_2^2$. These equations are easy to understand if we rotate the (orthogonal) vectors to form a canonical orthogonal basis $(a', 0, 0, \ldots), (0, b', 0, \ldots), (0, 0, c', \ldots)$, with $a', b', c' \in \mathbb{R}\backslash\{0\}$. Each gradient vector in the sum $S$ is now associated with a unique canonical orthogonal basis. Since the $L_2$ norm is invariant under basis rotations (for the same reason the length of a vector is invariant under rotation), it now becomes clear why subtracting a vector $v$ of the orthogonal basis from $S$ reduces the $L_2$ norm of $S$ (if $v$ is in the sum) or increases the sum's $L_2$ (if $v$ is not in the sum). Hence, our proposed attack computes $M(\boldsymbol{x}', c) = \mathbb{1}\{\|\Delta\Gamma_c^{(t+1)}\|_2^2 - \|\Delta\Gamma_c^{(t+1)} - \sum_{a \in \mathbb{Y}} \nabla_{\boldsymbol{W}}\mathcal{L}(a, \boldsymbol{x}'; \boldsymbol{W}_S^{(t)})\|_2^2 > 0\}$ as an indicator of membership of $\boldsymbol{x}'$ in the private data $\mathcal{D}_c$.

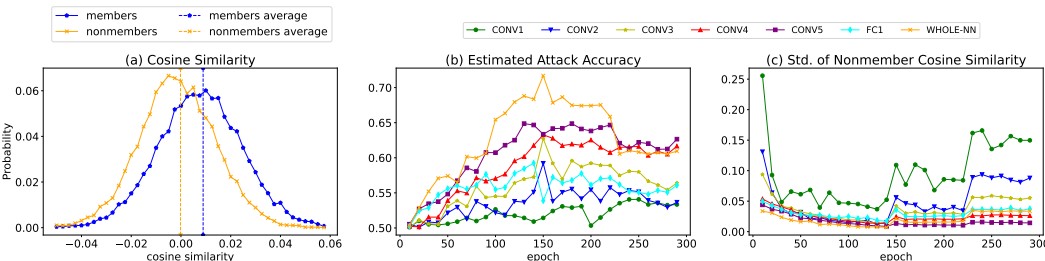

Figure 2: (a) Distribution of $\text{cosim}(\nabla_{\boldsymbol{W}}\mathcal{L}(a, \boldsymbol{x}'; \boldsymbol{W}_S^{(t)}), \Delta\Gamma_c^{(t+1)})$ for members and non-members, CIFAR-100 dataset, AlexNet, epoch 250, 5th convolutional layer. (b) Estimated attack accuracy for different layers as a function of averaged communication rounds (epochs). (c) Standard deviation of non-member cosine similarity distribution for different CNN layers (lower means it is easier to identify membership) as a function of averaged communication rounds (epochs).

**Attacks using multiple communication rounds:** In federated learning, we could also obtain $\Delta\Gamma_c^{(t)}$ from multiple communication rounds $t$. We would like to use multiple communication rounds to improve our attack effectiveness. In what follows we use the cosine similarity feature as an example to show how to combine attack features from different communication rounds. The attacker's goal is to use instances $\boldsymbol{x}'$ known to be non-members and estimate a cosine similarity distribution $\phi(\mu, \sigma)$

for them. Our empirical results for *a single round* in Figure 2(a), indicate that the cosine similarity distribution of non-members $\phi$ for a single communication round is distinct from members and approximately Gaussian. In order to increase separation between the two distributions in Figure 2(a), we average the cosine similarity over multiple communication rounds (epochs), see an exemplar result in Figure 9 in the Appendix. Figure 2(c) shows that, for most layers, using multiple rounds decreases the standard deviation, which in turn creates better separation between member and non-member distributions, which then significantly increases the effectiveness of our attack Figure 2(b). Moreover, Figures 4 and 5 in the Appendix shows that across communication rounds $t$, the standard deviation and averages of this Gaussian over a single round tends to remain the same, which means that there is no single communication round that would outperform using multiple communication rounds.

**Hypothesis test without member and non-member data:** Due to the Gaussianity of the cosine similarity data, we can design a membership detection approach without the need for non-member and member data using a Student's t-test (where the null hypothesis is that the cosine similarity is zero). This, however, tends to underestimate the false positive rate, since even with a jacknife estimate of the variance, the samples over consecutive communication rounds are not independent (as the neural network parameters are similar). In order to get a calibrated estimate of the false positive rate of our membership inference approach, we will use limited non-member data through a conformal hypothesis test.

**Conformal hypothesis test using non-member data:** Using limited non-member data, we can build an empirical distribution of the multi-round average cosine similarity (or gradient-diff metric) for non-members. Using this empirical distribution, we can set $\gamma$ of Equation (4) as the threshold of a conformal region corresponding to a desired false positive rate (Shafer & Vovk, 2008). This conformal hypothesis test has the advantage of being calibrated. This is how we set $\gamma$ in our experiments.

### 2.4 PRACTICAL IMPROVEMENTS TO PROPOSED PASSIVE ATTACK

In a large overparameterized neural network, using gradients of all parameters for our attack can be computationally expensive. To reduce this cost, we will focus on a single neural network layer to launch our MI attack. Note that the distributions of the cosine similarity in different layers are significantly different, leading to different effectiveness of membership inference (see Figure 2(c) to compare the cosine similarity standard deviations across layers, lower is better). Thus, we need to answer the question of how an attacker finds the optimal layer for the highest membership inference effectiveness.

In order to verify whether the approach of choosing the layer with the least cosine similarity standard deviation, we evaluate our attack accuracy (over a single communication round) as a function of the layer and the number of training epochs in Figure 2(b). We see that the 5th convolutional layer (the purple curve with square marker) has the highest attack accuracy as shown in Figure 2(b) while also having the lowest standard deviation as shown in Figure 2(c). For other datasets, we also observe the same phenomenon: The layer with the highest attack accuracy has the lowest standard deviation of cosine similarity distribution of non-members. Therefore, if the attacker only knows some non-members, the attacker can use the standard deviation of cosine similarity distribution of non-members as a rule of thumb to choose a layer to perform MI attacks.

### 2.5 EXISTING APPLICABLE DEFENSES AGAINST MEMBERSHIP INFERENCE ATTACK IN FEDERATED LEARNING

Differential privacy (Dwork, 2008; Dwork et al., 2006; Dwork, 2006) is a widely adopted privacy-preserving technique. A randomized algorithm $\mathcal{M}$ can satisfy $(\epsilon, \delta)$-differential privacy if given any two datasets $\mathcal{D}$ and $\mathcal{D}'$ that differ in at most one instance, for any output space $R$, the following equation holds:

$$\mathbb{P}(\mathcal{M}(\mathcal{D}) \in R) \leq \exp(\epsilon)\mathbb{P}(\mathcal{M}(\mathcal{D}') \in R) + \delta. \tag{5}$$

DP-SGD (Abadi et al., 2016) adds noise to the gradients (or other components) during the training process. Training with differential privacy provides theoretical and worst-case guarantee against any MI attacks. Achieving a meaningful theoretical guarantee (i.e., with a reasonably small $\epsilon$, e.g., $< 5$) requires the usage of very large noises, resulting significant testing accuracy drop. However, one

could use much smaller noises in DP-SGD (which results in very large $\epsilon$ in the theoretical guarantee) to provide empirical defense against MI attacks (Li et al., 2020). DP-SGD can be implemented in different level under federated learning setting: instance-level or client-level. Instance-level DP means that the noise is added to every step of local computation for each client and client-level DP means that the noise is added only when the central server is summing up the gradient update from all the clients. Naseri et al. (2020) evaluated how differential privacy would affect the federated learning process in both central DP and local DP settings. The experimental results show that if a meaningful privacy budget ($\epsilon < 10$) needs to be achieved, the utility cost is significant for both central DP and local DP (more than $10\%$ for CIFAR-100 dataset with AlexNet). In this paper, we consider the instance-level DP-SGD and we tune the parameters of DP-SGD so that the validation accuracy drop is less than $1\%$ to see if DP-SGD can still achieve empirical defense.

To reduce the communication overhead between clients and the central server in federated learning, Bernstein et al. (2018) proposed to use the sign of gradients instead of the numerical values of the gradients for transmission. In our case, since only the gradient signs are transmitted, the effectiveness of MI attacks from an eavesdropper should be reduced.

Li et al. (2020) proposed Mix-up+MMD defense to reduce the effectiveness of membership inference attacks in blackbox setting by closing the gap between training accuracy and testing accuracy. Zhang et al. (2017) proposed Mix-up training to achieve better generalization on large over-parameterized neural networks. Mix-up training uses linear interpolation of two different training instances to generate a mixed instance and train the classifier with the mixed instance. Since only the mixed instances are used in the training, the training accuracy on original training instances are expected to be reduced. On the other hand, Maximum Mean Discrepancy (MMD) (Fortet & Mourier, 1953; Gretton et al., 2012) is used to measure the difference between distribution of label confidence $F_\theta(x)_y$ for members and non-members. The measured difference is used as a regularization in training, to reduce the generalization gap. In Section 4 we test these above defenses against our proposed attack.

## 3 RELATED WORK

To the best of our knowledge, there is no specific existing attack designed for our setting. The existing attacks that can be adapted to our setting (**baselines**) are the blackbox attack using the prediction loss proposed in Yeom et al. (2018), the Gaussian noise attack called Merlin from Jayaraman et al. (2020) and the modified prediction entropy attack from Song et al. (2019). Due to space limitations and the extensive literature related to membership inference attacks and federated learning attacks (less so on the intersection), our extensive related work description can be found in Appendix A.9, including existing blackbox membership inference attacks and attacks in federated learning that need access to private members.

| | Skin AlexNet | Retina AlexNet | CIFAR-100 AlexNet | CIFAR-100 DenseNet |
|---|---|---|---|---|
| | PLR ↑ (FPR @ 2%) | PLR ↑ (FPR @ 1%) | PLR ↑ (FPR @ 1%) | PLR ↑ (FPR @ 1%) |
| blackbox loss (Yeom et al., 2018) | 1.39±0.39 | 1.06±0.49 | 1.02±0.39 | 1.08±0.34 |
| blackbox modified entropy (Song et al., 2019) | 1.24±0.32 | 0.94±0.33 | 1.06±0.34 | 1.03±0.40 |
| Merlin (Jayaraman et al., 2020) | 0.00±0.00 | 0.00±0.00 | 1.01±0.45 | 0.98±0.39 |
| **fed-loss attack** | 2.82±1.37 | 1.03±0.54 | 1.28±0.45 | 0.94±0.33 |
| **cosine attack** | 11.60±3.30 | 6.10±1.63 | **7.26**±1.54 | **3.22**±1.19 |
| **gradient-diff attack** | **16.79**±3.98 | **8.59**±1.29 | 4.48±1.21 | 1.82±0.66 |

Table 1: Evaluation of our cosine attack, gradient-diff attack and the blackbox loss attack without private information on hard medical image datasets and benchmark image dataset CIFAR-100. Results are averaged over 10 participants.

| | Skin InceptionV3 | Retina InceptionV3 | CIFAR-100 InceptionV3 |
|---|---|---|---|
| | PLR ↑ (FPR @ 2%) | PLR ↑ (FPR @ 1%) | PLR ↑ (FPR @ 1%) |
| blackbox loss (Yeom et al., 2018) | 1.40±0.34 | 1.19±0.44 | 1.07±0.45 |
| blackbox modified entropy (Song et al., 2019) | 1.39±0.34 | 1.41±0.28 | 1.29±0.36 |
| Merlin (Jayaraman et al., 2020) | 1.17±0.33 | 0.00±0.00 | 0.00±0.00 |
| **fed-loss attack** | 1.18±0.35 | 1.08±0.49 | 1.16±0.35 |
| **cosine attack** | 8.55±4.25 | 1.79±0.42 | 2.02±0.43 |
| **gradient-diff attack** | **10.00**±3.75 | **2.30**±0.92 | **4.64**±1.36 |

Table 2: Evaluation of our cosine attack, gradient-diff attack and the blackbox loss attack without private information on hard medical image datasets and benchmark image dataset CIFAR-100. Results are averaged over 10 participants. Transfer learning using InceptionV3 pretrained on Imagenet.

## 4 RESULTS

We now evaluate our proposed attack against baselines in two hard medical image tasks (Retina and Skin), one benchmark image dataset (CIFAR-100), two datasets with only binary features (Purchase and Texas), and three easy medical image task (Medical-MNIST, Pneumonia and Kidney) (the results in the easy tasks are in the Appendix). Detailed description on datasets is included in Appendix A.1. A detailed description of the training parameters is given in Table 6 in the Appendix.

In our federated learning setting, we consider 10 clients that are included in every round of training. Furthermore, we assume that all clients communicate with the central server once after one local epoch (equivalent to a few gradient steps). For instance, for CIFAR-100 as each client has 4000 instances, which means that at 100 image mini-batches, each client performs 40 steps of SGD (with mini-batch size 100) and then sends its parameter update to the central server, when the attacker eavesdrop and gets access to a model update sent to the server by a specific client.

For attack settings, we assume that the attacker does not know any member used in the training. We also assume that the attacker is able to observe model updates at every round of communication. Furthermore, the attacker is assumed to have a small validation set (which contains only non-members). Besides the baselines described in Section 3, one further baseline we consider is to use loss from all communication rounds and we call this baseline the **fed-loss** baseline. To combine information from different communication rounds, we use the average of the attack metric (e.g. cosine similarity) from all collected rounds to produce a single number for evaluation.

To evaluate the membership inference attack against each participant, following existing literature (Nasr et al., 2019; Shokri et al., 2017; Carlini et al., 2021), we create a balanced evaluation set (half members and half non-members). For Pneumonia, Skin and Kidney datasets, due to lack of data, we can only create a balanced evaluation set with size of 400, 1000, 800 respectively. For remaining datasets, the size of the evaluation set is fixed as 2000 (1000 members and 1000 non-members). Following Carlini et al. (2021), we use the Positive Likelihood Ratio (PLR) at a low False Positive Rate (FPR) to evaluate different attacks. PLR values indicate the rate between true positive and false positive examples: Values larger than one indicate a successful attack that is able to generate more true positive than false positive cases. In our evaluation, we use TPR at the threshold necessary to get $1\%$ false positive rate for Medical-MNIST, Retina, CIFAR-100 and Purchase. For the remaining medical image datasets (Pneumonia, Skin and Kidney), due to lack of sufficient data, we use $5\%, 2\%, 2.5\%$ FPR (defined as enough to provide at least ten false positive examples in the data). All these FPR thresholds are chosen to reflect how many correct members can be detected before the attacker meets the first false positive example. We also use AUC (area under curve) to evaluate all attacks to show the average-case performance, described in Table 11 in the Appendix.

*Attacks to transfer learning.* Transfer learning is commonly used in medical image tasks. Following Minaee et al. (2020), we study the transfer learning case where only the last fully-connected layer is further fine-tuned on the medical image dataset. We mainly study two popular pretrained models in this application: InceptionV3 and Resnet50. The hyperparameter ranges used in training are described in Tables 7 and 10 of the Appendix.

**Attack on hard medical image datasets and benchmark image dataset.** The PLR results at low FPR for Skin, Retina, and CIFAR-100 are summarized in Table 1, showing that both the cosine and gradient-diff attacks are highly effective for AlexNet, with the cosine attack being more effective for DenseNet than the gradient-diff attack. The baselines and the naive fed-loss attack are not nearly as effective. The attacks on federated transfer learning against InceptionV3 and Resnet50 are shown in Tables 2 and 4, respectively. Again we see that that cosine and gradient-diff attacks are highly effective for Skin and reasonably effective for Retina and CIFAR-100, all significantly better than the baseline and fed-loss attacks.

**Attack on datasets with only binary features.** Table 5 presents the PRL at low FPR of attacks on Purchase and Texas datasets, showing really good effectiveness of the cosine attack, while gradient-diff attack is ineffective at Purchase but effective at Texas. All baseline and fed-loss attacks are ineffective.

**Defense evaluation against our (orthogonal gradient) attackers.** We now evaluate the DP-SGD, SIGN-SGD, and Mix-up+MMD defenses against our two proposed attacks (cosine and gradient-diff attacks). For DP-SGD defense, we tune the hyperparameters of DP-SGD and SIGN-SGD so that the

Figure 3: We change the level of overparameterization of the target neural networks by varying the rank of the last fully connected layer and measure the attack effectiveness of our proposed attack.

| | Skin AlexNet | Retina AlexNet | CIFAR-100 AlexNet | CIFAR-100 DenseNet | Purchase FC-NN | Texas FC-NN |
|---|---|---|---|---|---|---|
| | PLR ↑ (FPR @ 2%) | PLR ↑ (FPR @ 1%) | PLR ↑ (FPR @ 1%) | PLR ↑ (FPR @ 1%) | PLR ↑ (FPR @ 1%) | PLR ↑ (FPR @ 1%) |
| **cosine attack - NO-DEF** | 11.60±3.30 | 6.10±1.63 | 7.26±1.54 | 3.22±1.19 | 11.09±1.97 | 25.45±3.42 |
| **cosine attack - DP-SGD** | 38.31±1.92 | 6.86±1.65 | 5.07±1.94 | N/A | 10.21±1.40 | 19.28±2.38 |
| **cosine attack - Sign-SGD** | 2.74±0.55 | 7.01±1.52 | 1.82±0.48 | 2.27±0.60 | 10.19±2.46 | N/A |
| **cosine attack - Mix-up+MMD** | N/A | 4.13±1.25 | 2.96±0.69 | N/A | N/A | N/A |
| **gradient-diff attack - NO-DEF** | 16.79±3.98 | 8.59±1.29 | 4.48±1.21 | 1.82±0.66 | 0.85±0.40 | 6.96±1.30 |
| **gradient-diff attack - DP-SGD** | 37.12±3.21 | 8.96±2.15 | 3.49±0.71 | N/A | 1.30±0.36 | 3.92±0.81 |
| **gradient-diff attack - Sign-SGD** | 2.98±0.72 | 7.51±1.24 | 1.94±0.57 | 2.09±0.81 | 1.81±0.62 | N/A |
| **gradient-diff attack - Mix-up+MMD** | N/A | 5.55±1.25 | 3.57±0.90 | N/A | N/A | N/A |

Table 3: Evaluation of DP-SGD and Sign-SGD defenses against our proposed attacks. Results are averaged over 10 participants. Note that Mix-up+MMD is not applicable to binary feature datasets such as Purchase and Texas, since Mix-up on binary datasets would lead to training failure. Results marked with N/A are not available due to training difficulties we have not been able to circumvent.

validation accuracy drop is less than $1\%$. Hyperparameters used in DP-SGD defense are described in Table 17 in the appendix. The results are summarized in Table 3, showing that Sign-SGD is the most effective defense against our attacks, but it is not enough to completely eliminate the risk. For instance, in the Retina dataset there was no defense that could significantly mitigate the attack risk.

**Relationship between the level of model overparameterization and attack effectiveness.** Our work is predicated on the hypothesis that our gradient orthogonality attacks are effective because of model overparameterization. In this experiment we use the capacity control procedure in Martin & Mahoney (2021) to adjust model overparameterization without changing the number of neurons. This is done by defining the weight matrix of the only fully connected layer as a the product of two rank $K$ matrices, so that the weight matrix has rank $K$. Reducing $K$ reduces the overparameterization of the model. Indeed, Figure 3(a-b) shows that the higher the $K$, the more effective our gradient orthogonality attacks become, which points to the potential validity of our hypothesis.

However, as $K$ increases so does the generalization gap, which could be an alternative reason why our attack is successful. Hence, in order to validate the hypothesis that overparameterization (not generalization gap) influences the effectiveness of our attack, we apply the Mix-up+MMD defense to the results in Figure 3(c-d), which practically eliminates the generalization gap, but still shows that our attack effectiveness approximately increases according to $\sqrt{K}$ with the rank $K$ (the level of overparameterization of the neural network model), validating our hypothesis.

## 5 CONCLUSIONS

This work considered the challenge of performing membership inference attacks in a federated learning setting —for image classification— where an adversary can only observe the communication between the central node and a single client (a passive white-box attack). The key insight of our method is the observation that the gradient of large overparameterized neural network models statistically behave like high-dimensional independent isotropic random vectors at later stages of the optimization. Using this insight, we devised two attacks that are often little impacted by existing and proposed defenses. Finally, we validated the hypothesis that our attack depends on the overparameterization by showing that increasing the level of overparameterization (without changing the neural network architecture) positively correlates with our attack effectiveness.

## 6 ACKNOWLEDGEMENTS

This work was funded in part by the National Science Foundation (NSF) Awards CAREER IIS-1943364 and CCF1918483. Any opinions, findings and conclusions or recommendations expressed in this material are those of the authors and do not necessarily reflect the views of the sponsors.

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

# A    APPENDIX

|  | Skin
Resnet50 | Retina
Resnet50 | CIFAR-100
Resnet50 |
| --- | --- | --- | --- |
|  | PLR ↑
(FPR @ 2%) | PLR ↑
(FPR @ 1%) | PLR ↑
(FPR @ 1%) |
| blackbox loss (Yeom et al., 2018) | 1.01±0.22 | 1.02±0.18 | 1.10±0.40 |
| blackbox modified entropy (Song et al., 2019) | 1.02±0.28 | 1.05±0.24 | 1.18±0.36 |
| Merlin (Jayaraman et al., 2020) | 0.00±0.00 | 0.00±0.00 | 0.00±0.00 |
| **fed-loss attack** | 1.24±0.44 | 0.85±0.25 | 1.13±0.65 |
| **cosine attack** | 9.25±4.25 | 2.09±0.51 | 2.03±0.49 |
| **gradient-diff attack** | **9.53**±2.61 | **2.39**±0.76 | **3.66**±1.24 |

Table 4: Evaluation of our cosine attack, gradient-diff attack and the blackbox loss attack without private information on hard medical image datasets and benchmark image dataset CIFAR-100. Results are averaged over 10 participants. Transfer learning using Resnet50 pretrained on Imagenet.

|  | Purchase
FC-NN | Texas
FC-NN |
| --- | --- | --- |
|  | PLR ↑
(FPR @ 1%) | PLR ↑
(FPR @ 1%) |
| blackbox loss (Yeom et al., 2018) | 1.04±0.64 | 1.23±0.31 |
| blackbox modified entropy (Song et al., 2019) | 0.96±0.31 | 0.92±0.30 |
| Merlin (Jayaraman et al., 2020) | 0.98±0.46 | 0.99±0.38 |
| **fed-loss attack** | 1.25±0.57 | 1.64±0.87 |
| **cosine attack** | **11.09**±1.97 | **25.45**±3.82 |
| **gradient-diff attack** | 0.85±0.40 | 6.96±1.30 |

Table 5: Evaluation of our cosine attack, gradient-diff attack and the blackbox loss attack without private information on binary feature datasets. Results are averaged over 10 participants.

## A.1    DATASET DESCRIPTION

The medical-MNIST dataset apolanco3225 (2017) is a simple MNIST-style medical images in 64x64 dimension; There were originally taken from other datasets and processed into such style. There are 58,954 medical images belonging to 6 classes. We sampled a balanced data set with 53,724 images, 8954 images in each class. We divide this dataset into 3 disjoint set: 40,000 images for training, 5,000 images for validation and 8,724 images for testing. We resize the images to $32 \times 32$ uniformly. We change the input shape of the first linear layer of LeNet to fit the input image size.

A pneumonia dataset was proposed in Kermany et al. (2018). This pneumonia dataset contains 5,863 X-Ray images in 2 categories (Pneumonia/Normal). Chest X-ray images (anterior-posterior) were selected from retrospective cohorts of pediatric patients of one to five years old from Guangzhou Women and Children's Medical Center, Guangzhou. All chest X-ray imaging was performed as part of patients' routine clinical care. For the analysis of chest x-ray images, all chest radiographs were initially screened for quality control by removing all low quality or unreadable scans. The diagnoses for the images were then graded by two expert physicians before being cleared for training the AI system. In order to account for any grading errors, the evaluation set was also checked by a third expert. We sampled a balanced data set with 2,698 images. We divide this dataset into 3 disjoint set: 2,000 images for training, 200 images for validation and 498 images for testing. We resize X-Ray images to $64 \times 64$ uniformly because original images are in different sizes.

Authors of Kermany et al. (2018) also proposed the Retina dataset contains retinal optical coherence tomography images to capture high-resolution cross sections of the retinas of living patients. This dataset consists of 84,495 X-Ray images. This dataset is divided into a training set of 83,484 images and a testing set of 968 images. There are four classes in this dataset: CNV, DME, DRUSEN and NORMAL. The far left image is to show choroidal neovascularization (CNV) with neovascular membrane (white arrowheads) and associated subretinal fluid (arrows). The middle left image is to show diabetic macular edema (DME) with retinal-thickening-associated intraretinal fluid (arrows). The middle right image is to show multiple drusen (arrowheads) present in early AMD. The far right image is to show normal retina with preserved foveal contour and absence of any retinal fluid/edema. We sampled a balanced training set with 34,464 images. The given testing set is already balanced. We resize images to $64 \times 64$ uniformly because original images are in different sizes.

CT Kidney dataset Islam et al. (2022) was collected from PACS (Picture archiving and communication system) from different hospitals in Dhaka, Bangladesh where patients were already diagnosed with having a kidney tumor, cyst, normal or stone findings. Both the Coronal and Axial cuts were selected from both contrast and non-contrast studies with protocol for the whole abdomen and urogram. The Dicom study was then carefully selected, one diagnosis at a time, and from those we created a batch of Dicom images of the region of interest for each radiological finding. Following that, we excluded each patient's information and meta data from the Dicom images and converted the Dicom images to a lossless jpg image format. After the conversion, each image finding was again verified by a radiologist and a medical technologist to reconfirm the correctness of the data. We sampled a balanced data set with 5,508 images. We divide this dataset into 3 disjoint set: 4,000 images for training, 500 images for validation and 1,008 images for testing. We resize images to $64 \times 64$ uniformly because original images are in different sizes.

The skin disease dataset is taken from the public portal Dermnet (`http://www.dermnet.com/`) which is the largest dermatology source online built for the purpose of providing online medical education. The data consists of images of 23 types of skin diseases and around 19,500 images. We sampled a balanced data set with 6,095 images. We divide this dataset into 3 disjoint set: 5,000 images for training, 500 images for validation and 595 images for testing. We resize images to $64 \times 64$ uniformly because original images are in different sizes. This dataset is available at `https://www.kaggle.com/datasets/shubhamgoel27/dermnet`.

Wang et al. (2020) presented an open-access benchmark covid dataset that contains 30384 chest x-ray(CXR) images. This dataset is a collection from several publicly available data sources (Cohen et al., 2020; Chowdhury et al., 2020; Rahman et al., 2021; Wang et al., 2017; Tsai et al., 2021b;a; Clark et al., 2013; Saltz et al., 2021; Clark et al., 2013). This is a binary classification dataset, which has 14191 covid-negative images and 16191 covid-positive images. We sampled a balanced dataset with 28382 images (14191 images for each class). We divide this balanced dataset into 3 disjoint set: 20000 images for training, 4000 images for validation and 4382 images for test. We resize images to $64 \times 64$ uniformly because original images are in different sizes.

CIFAR Krizhevsky et al. (2009) dataset is one benchmark dataset for evaluating image classification algorithms. They contain 60,000 color images of size $32 \times 32$, divided into 50,000 for training and 10,000 for testing. In CIFAR-100, these images are divided into 100 classes, with 600 images for each class. In CIFAR-10, these 100 classes are grouped into 10 more coarse-grained classes; there are thus 6000 images for each class. These two datasets are widely used to evaluate membership inference attack in Shokri et al. (2017); Salem et al. (2019); Nasr et al. (2018; 2019); Yeom et al. (2018). For CIFAR-10 and CIFAR-100 dataset, we use AlexNet and DenseNet-BC(100,12) as the target model, as they are the standard neural networks used in previous paper Nasr et al. (2019).

The Purchase dataset is based on the "acquire valued shopper" challenge from Kaggle. This dataset includes shopping records for several thousand individuals. We obtained the processed and simplified version of this dataset from the authors of Shokri et al. (2017). Each data instance has 600 binary features. This dataset is clustered into 100 classes and the task is to predict the class for each customer. The dataset contains 197,324 data instances. This dataset is also widely used to evaluate membership inference attack in Shokri et al. (2017); Salem et al. (2019); Nasr et al. (2018; 2019); Jia et al. (2019). We use the fully connected neural network from Nasr et al. (2019) for this Purchase dataset.

The Texas dataset includes hospital discharge data. The records in the dataset contain information about inpatient stays in several health care facilities published by the Texas Department of State Health Services. Data records have features about the external causes of injury, the diagnosis, the procedures the patient underwent, and generic information. We obtained a processed version of the dataset from Shokri et al. (2017). This dataset contains 67,330 records and 6,170 binary features which represent the 100 most frequent medical procedures. The records are clustered into 100 classes, each representing a different type of patient. This dataset is used to evaluate membership inference attack in (Shokri et al., 2017; Nasr et al., 2018; 2019; Jia et al., 2019).

## A.2 COSINE SIMILARITY BETWEEN CLIENT UPDATES AND GRADIENTS OF INSTANCES

We provide the cosine similarity between gradients of one instance and client update across all epochs for each dataset. Our goal is to determine a starting epoch to launch our proposed attack. In

| Dataset | Medical-MNIST | Pneumonia | Kidney | covid | Skin | Retina | CIFAR-100 | CIFAR-100 | Purchase | Texas |
|---|---|---|---|---|---|---|---|---|---|---|
| Model | LeNet | AlexNet | AlexNet | AlexNet | AlexNet | AlexNet | AlexNet | DenseNet-BC(100,12) | 4-layer-FC-NN | 4-layer-FC-NN |
| Epoch | 50 | 150 | 150 | 100 | 150 | 100 | 300 | | 100 | 100 |
| Optimizer | | | ← | | SGD, momentum 0.9 | | → | | Adam | Adam |
| Initial learning rate | 0.01 | 0.01 | 0.02 | 0.01 | 0.1 | 0.1 | 0.2 | 0.2 | 0.001 | 0.001 |
| Scheduling epochs | None | None | None | None | None | None | [150,225] | [250,375] | None | None |
| Batch size | 100 | 100 | 100 | 100 | 100 | 100 | 100 | 100 | 100 | 100 |
| Number of clients | 10 | 10 | 10 | 10 | 10 | 10 | 10 | 10 | 10 | 10 |
| Number of classes | 6 | 2 | 4 | 2 | 23 | 4 | 100 | 100 | 100 | 100 |
| weight decay | 1e-5 | 1e-5 | 1e-5 | 1e-5 | 1e-5 | 1e-5 | 1e-5 | 1e-5 | 1e-5 | 1e-5 |
| Data augmentation | None | None | None | None | None | None | ← | crop & horizontal-flip → | None | None |
| Total number of data | 53724 | 2698 | 5508 | 28382 | 6095 | 34464 | 60000 | 60000 | 197324 | 67330 |
| Training set size for one client | 4000 | 200 | 400 | 2000 | 500 | 3000 | 4000 | 4000 | 4000 | 4000 |
| Validation set size | 5000 | 200 | 500 | 4000 | 500 | 10000 | 10000 | 10000 | 10000 | 10000 |
| Testing set size | 8724 | 498 | 1008 | 4382 | 595 | 2464 | 10000 | 10000 | 147324 | 17330 |
| Balanced MI evaluation set size per client | 2000 | 400 | 800 | 2000 | 1000 | 2000 | 2000 | 2000 | 2000 | 2000 |
| FPR threshold for MI evaluation | 1% | 5% | 2.5% | 1% | 2% | 1% | 1% | 1% | 1% | 1% |

Table 6: Neural network architecture and hyperparameters for datasets evaluated in this paper. We mainly use LeNet from LeCun et al. (1998), AlexNet from Krizhevsky et al. (2012) and DenseNet from Huang et al. (2017). When current epoch is included in the list of scheduling epochs, the learning rate is multiplied by $0.1$. The 4-layer-FC-NN for the Purchase dataset and the Texas dataset contains the following 4 fully-connected layer: FC1 with 1024 neurons, FC2 with 512 neurons, FC3 with 256 neurons and FC4 with 100 neurons.

| Dataset | Pneumonia | Kidney | Covid | Skin | Retina | CIFAR-100 |
|---|---|---|---|---|---|---|
| Epoch | 100 | 100 | 100 | 100 | 100 | 100 |
| Optimizer | | ← | Adam | | → | |
| Initial learning rate | 0.01 | 0.01 | 0.1 | 0.001 | 0.001 | 0.0001 |
| Batch size | 100 | 100 | 100 | 100 | 100 | 100 |
| Number of clients | 10 | 10 | 10 | 10 | 10 | 10 |
| Number of classes | 2 | 4 | 2 | 23 | 4 | 100 |
| weight decay | 1e-5 | 1e-5 | 1e-5 | 1e-5 | 1e-5 | 1e-5 |
| Data augmentation | ← | | center-crop & normalization | | | → |
| Total number of data | 2698 | 5508 | 28382 | 6095 | 34464 | 60000 |
| Training set size for one client | 200 | 400 | 2000 | 500 | 3000 | 4000 |
| Validation set size | 200 | 500 | 4000 | 500 | 2000 | 10000 |
| Testing set size | 498 | 1008 | 4382 | 595 | 2464 | 10000 |
| Balanced MI evaluation set size per client | 400 | 800 | 2000 | 1000 | 2000 | 2000 |
| FPR threshold for MI evaluation | 5% | 2.5% | 1% | 2% | 1% | 1% |
| Training accuracy | 0.966 | 0.956 | | 0.485 | 0.786 | 0.553 |
| Testing accuracy | 0.916 | 0.929 | | 0.290 | 0.761 | 0.511 |

Table 7: Hyperparameters for transfer learning using InceptionV3 pretrained on Imagenet.

figure 4, we show the cosine similarity distributions for hard image datasets and CIFAR-100 dataset. In the first stage, the mean cosine similarity for non-members starts with a non-zero number and lasts for a few epochs. In the second stage, the mean of cosine similarity for non-members goes to zero and fluctuate around zero. Our proposed attacks exploit the observation that the cosine similarity between non-members and client updates is close to zero after several epochs, which is observed in the second stage. Thus, our proposed attacks should start at the second stage. For the four datasets considered here, we choose to start our proposed attack at epoch 40,40,100,150 for retina, skin, CIFAR-100 with AlexNet, CIFAR-100 with DenseNet respectively.

In Figure 5, we show the cosine similarity distributions for easy medical image datasets. We can conclude that the standard deviations of these two distributions (one for member and one for non-member) are very similar, however the mean of these two distributions are sometimes different. Thus, for easy medical image datasets, we choose to collect cosine similarities from the first epoch.

Overall, from the attacker's perspective, the attacker could look at the mean of cosine similarity between non-members and client updates. If there is a noticeable drop, as shown in Figure 4, then the attack should start after the mean goes to zero. Otherwise it is better to use the attack related information from all epochs.

A.3 ATTACKS WITH PRIVATE INFORMATION

Even though comparing an attack that makes use of private information $\mathcal{D}_c$ against our attack which does not use private information is an unfair, we still found the results interesting. Zari's attack is similar to the fed-loss baseline we described earlier, with extra knowledge of known members (private information) and known non-members. We implemented the Zari's attack by ourselves

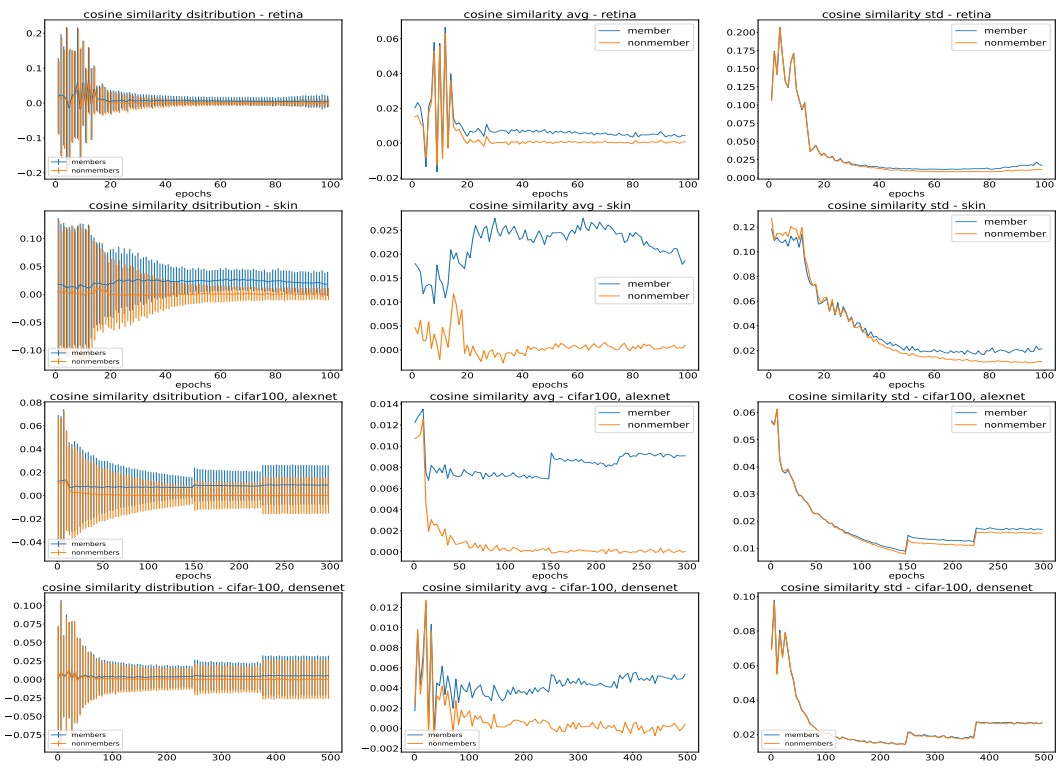

Figure 4: Cosine similarity distribution for hard medical image datasets (Skin & Retina) and CIFAR-100 dataset.

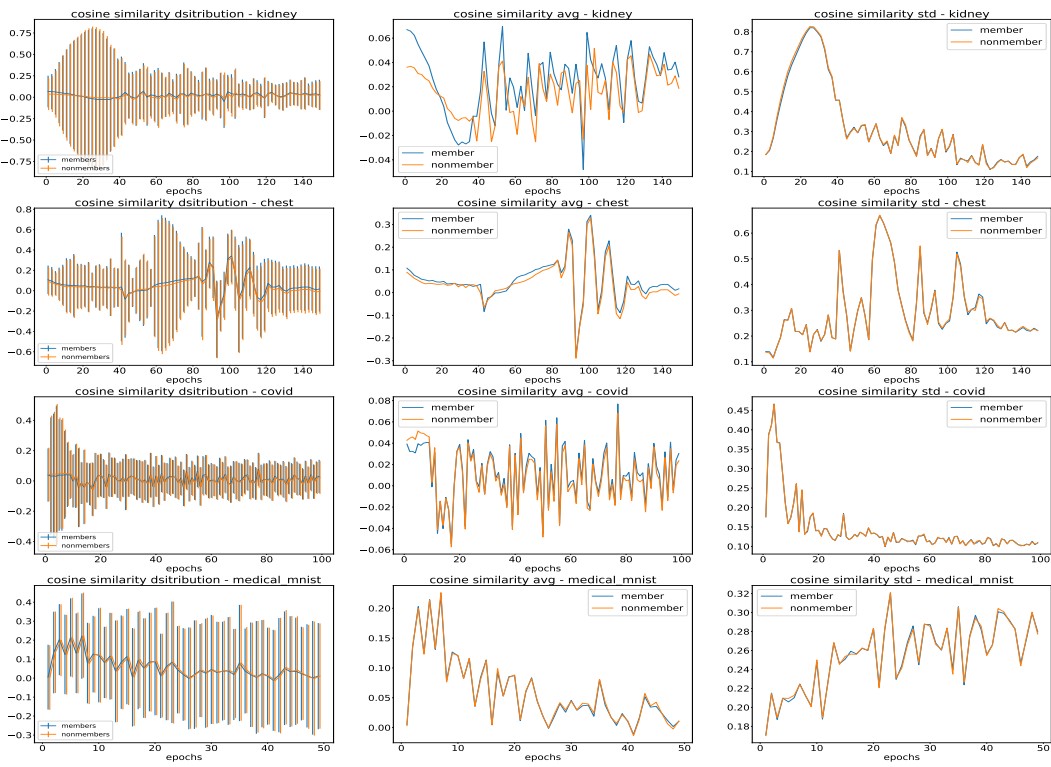

Figure 5: Cosine similarity distribution for easy medical image datasets.

| | Medical-MNIST LeNet | Pneumonia AlexNet | Kidney AlexNet | Covid AlexNet |
|---|---|---|---|---|
| | PLR ↑ (FPR @ 1%) | PLR ↑ (FPR @ 5%) | PLR ↑ (FPR @ 2.5%) | PLR ↑ (FPR @ 1%) |
| blackbox loss (Yeom et al., 2018) | 1.25±0.53 | 0.94±0.39 | 0.88±0.43 | 1.20±0.43 |
| blackbox modified entropy (Song et al., 2019) | 0.73±0.44 | 0.95±0.50 | 1.04±0.47 | 1.33±0.35 |
| Merlin (Jayaraman et al., 2020) | 0.00±0.00 | 0.00±0.00 | 0.00±0.00 | 0.00±0.00 |
| **fed-loss attack** | 0.99±0.49 | 0.96±0.52 | 1.61±0.32 | 0.84±0.38 |
| **cosine attack** | 1.31±0.56 | 1.33±0.37 | 2.05±0.55 | 2.25±1.08 |
| **gradient-diff attack** | 1.51±0.36 | 1.64±0.48 | 2.16±0.61 | 4.18±1.28 |

Table 8: Evaluation of our cosine attack, gradient-diff attack and the blackbox loss attack without private information on easy medical image datasets. Results are averaged over 10 participants.

| | Pneumonia InceptionV3 | Kidney InceptionV3 | Covid InceptionV3 |
|---|---|---|---|
| | PLR ↑ (FPR @ 5%) | PLR ↑ (FPR @ 2.5%) | PLR ↑ (FPR @ 1%) |
| blackbox loss (Yeom et al., 2018) | 1.12±0.54 | 1.04±0.39 | 1.03±0.33 |
| blackbox modified entropy (Song et al., 2019) | 0.88±0.38 | 1.10±0.54 | 0.98±0.40 |
| Merlin (Jayaraman et al., 2020) | 0.00±0.00 | 0.00±0.00 | 0.00±0.00 |
| **fed-loss attack** | 0.90±0.35 | 1.08±0.48 | 0.87±0.33 |
| **cosine attack** | 2.29±0.54 | 3.25±0.76 | 2.74±1.02 |
| **gradient-diff attack** | 1.30±0.64 | 2.83±0.52 | 2.42±0.71 |

Table 9: Evaluation of our cosine attack, gradient-diff attack and the blackbox loss attack without private information on easy medical image datasets. Results are averaged over 10 participants. Transfer learning using InceptionV3 pretrained on Imagenet.

since there is no public implementation. Moreover, we assume that the attacker is able to collect attack related information (e.g. loss) from every communication round.

The evaluation results are included in Table 12. For medical image datasets and Purchase dataset, our proposed attacks perform consistently better than Zari's attack. For CIFAR-100 dataset however, our proposed attacks perform a little worse than Zari's attack. It is surprising to see that our attacks (without private information) are superior in medical image datasets to Zari's attack (which uses private information).

### A.4 FEDERATED LEARNING V.S. BLACKBOX ATTACKS

We now switch gears and consider blackbox MI attacks on the final trained model. Surprisingly, we find that existing blackbox MI attacks on a federated-learned model are significantly less effective than the same attack over the same data with a model that was optimized with traditional centralized learning.

Examining this phenomenon further, we find that **FL-style training can be utilized as an effective defense against blackbox MI attacks** even when one has the whole training dataset and therefore does not need federated learning. More specifically, when given a training dataset $D$, one can conduct training in two different ways. Using the standard training method, in each epoch one goes through each minibatch while constantly updating the model. Using FL style training, one first partitions the training dataset, which can be viewed as creating virtual participants in FL. In each epoch one starts from one model, then trains on different partitions separately, and then average the resulting models.

We perform experiments while varying the number of virtual participants $K$ from 1 to 20. $D$, which contains 40000 training instances, is divided into $K$ equal-size, disjoint partitions, and each participant will perform training using one partition for one epoch, before we average the model. The results are summarized in Figure 6. Table 14 and Table 15 provide more detailed numbers for reference.

Moreover, it is not necessary that the model trainer uses the fixed data partitions created at the beginning of FL style training since all the clients are virtual. In Figure 6(c) and (d), we show the

| Dataset | Pneumonia | Kidney | Covid | Skin | Retina | CIFAR-100 |
|---|---|---|---|---|---|---|
| Epoch | 100 | 100 | 100 | 100 | 100 | 100 |
| Optimizer | | ← | Adam | | → | |
| Initial learning rate | 0.01 | 0.01 | 0.1 | 0.001 | 0.001 | 0.0001 |
| Batch size | 100 | 100 | 100 | 100 | 100 | 100 |
| Number of clients | 10 | 10 | 10 | 10 | 10 | 10 |
| Number of classes | 2 | 4 | 2 | 23 | 4 | 100 |
| weight decay | 1e-5 | 1e-5 | 1e-5 | 1e-5 | 1e-5 | 1e-5 |
| Data augmentation | ← | | center-crop & normalization | | | → |
| Total number of data | 2698 | 5508 | 28382 | 6095 | 34464 | 60000 |
| Training set size for one client | 200 | 400 | 2000 | 500 | 3000 | 4000 |
| Validation set size | 200 | 500 | 4000 | 500 | 2000 | 10000 |
| Testing set size | 498 | 1008 | 4382 | 595 | 2464 | 10000 |
| Balanced MI evaluation set size per client | 400 | 800 | 2000 | 1000 | 2000 | 2000 |
| FPR threshold for MI evaluation | 5% | 2.5% | 1% | 2% | 1% | 1% |
| Training accuracy | 0.978 | 0.975 | 0.962 | 0.501 | 0.801 | 0.642 |
| Testing accuracy | 0.959 | 0.945 | 0.953 | 0.324 | 0.776 | 0.594 |

Table 10: Hyperparameters for transfer learning using Resnet50 pretrained on Imagenet.

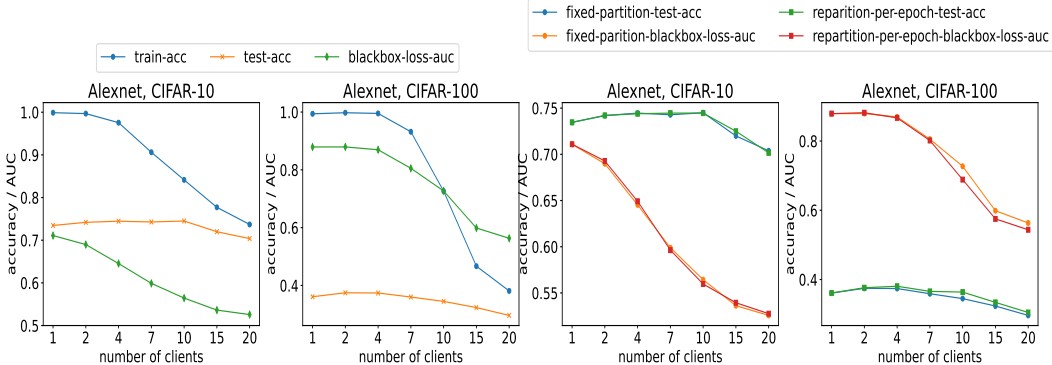

Figure 6: Comparing testing accuracy and blackbox loss AUC score between fixed data partitioning case and repartitioning-data-every-epoch case.

testing accuracy and the blackbox loss attack AUC for two cases: one is to use a fixed data partitions and another is to create new data partitions every epoch. For CIFAR-10 dataset and CIFAR-100 dataset with no more than 7 virtual clients, there is no obvious difference between two data partition strategies across different number of clients. For CIFAR-100 dataset with more than 7 clients, we can see that lower attack AUC and higher testing accuracy can be achieved by the re-partitioning step. It is worth noting that when the number of virtual clients is 10 and the re-partitioning is applied for CIFAR-100 dataset, the resulted testing accuracy is 0.28% higher than the testing accuracy from centralized training, while achieving 0.1906 blackbox MI AUC reduction.

In summary, what we have found suggests that one can utilize federated learning as a resilient defense against MI attacks by creating an adequate number of virtual clients in centralized setting. Since all clients are virtual, the model trainer could create different data partitions during training, which might help to achieve higher testing accuracy and lower blackbox MI attack AUC.

| | Pneumonia AlexNet | Kidney AlexNet | Covid AlexNet | Skin AlexNet | Retina AlexNet | CIFAR-100 AlexNet | CIFAR-100 DenseNet | Purchase FC-NN | Texas FC-NN |
|---|---|---|---|---|---|---|---|---|---|
| | AUC ↑ | AUC ↑ | AUC ↑ | AUC ↑ | AUC ↑ | AUC ↑ | AUC ↑ | AUC ↑ | AUC ↑ |
| **cosine attack** | 0.51±0.01 | 0.56±0.02 | 0.56±0.01 | 0.87±0.02 | 0.67±0.02 | 0.74±0.02 | 0.64±0.02 | 0.68±0.01 | 0.75±0.01 |
| **gradient-diff attack** | 0.51±0.01 | 0.57±0.02 | 0.57±0.01 | 0.88±0.01 | 0.71±0.02 | 0.74±0.02 | 0.64±0.02 | 0.62±0.01 | 0.70±0.01 |
| **fed-loss attack** | 0.50±0.01 | 0.52±0.02 | 0.51±0.01 | 0.85±0.01 | 0.59±0.01 | 0.68±0.02 | 0.61±0.02 | 0.61±0.01 | 0.63±0.01 |
| blackbox loss (Yeom et al., 2018) | 0.50±0.01 | 0.52±0.01 | 0.52±0.01 | 0.86±0.01 | 0.61±0.01 | 0.72±0.02 | 0.61±0.01 | 0.61±0.01 | 0.61±0.01 |

Table 11: AUC results for all datasets.

| | Pneumonia AlexNet | Kidney AlexNet | Skin AlexNet | Retina AlexNet | CIFAR-100 AlexNet | CIFAR-100 DenseNet | Purchase FC-NN |
|---|---|---|---|---|---|---|---|
| | PLR ↑ (FPR @ 10%) | PLR ↑ (FPR @ 5%) | PLR ↑ (FPR @ 4%) | PLR ↑ (FPR @ 2%) | PLR ↑ (FPR @ 2%) | PLR ↑ (FPR @ 2%) | PLR ↑ (FPR @ 2%) |
| Zari's attack (Zari et al., 2021) | 0.85 | 1.27 | 11.09 | 2.87 | 5.33 | 1.70 | 1.52 |
| **cosine attack** | 1.30 | 1.92 | 21.45 | 6.36 | 8.32 | 1.73 | 5.64 |
| **gradient-diff attack** | 0.80 | 1.44 | 24.22 | 2.92 | 2.21 | 1.32 | 0.91 |

Table 12: Evaluation of our cosine attack, gradient-diff attack and the Zari's attack private information. Results are averaged over 10 participants.

| | Pneumonia Resnet50 | Kidney Resnet50 | Covid Resnet50 |
|---|---|---|---|
| | PLR ↑ (FPR @ 5%) | PLR ↑ (FPR @ 2.5%) | PLR ↑ (FPR @ 1%) |
| blackbox loss (Yeom et al., 2018) | 1.06±0.26 | 1.26±0.35 | 1.09±0.49 |
| blackbox modified entropy (Song et al., 2019) | 0.99±0.34 | 1.30±0.32 | 0.99±0.31 |
| Merlin (Jayaraman et al., 2020) | 0.00±0.00 | 0.00±0.00 | 0.00±0.00 |
| **fed-loss attack** | 0.74±0.28 | 1.26±0.44 | 1.15±1.25 |
| **cosine attack** | 1.51±0.42 | 2.62±0.59 | 1.91±0.71 |
| **gradient-diff attack** | 1.43±0.60 | 2.36±0.69 | 2.14±0.79 |

Table 13: Evaluation of our cosine attack, gradient-diff attack and the blackbox loss attack without private information on easy medical image datasets. Results are averaged over 10 participants. Transfer learning using Resnet50 pretrained on imagenet.

## A.5 ABLATION STUDY ON NUMBER OF POINTS USED TO COMPUTE CLIENT UPDATES

In this subsection, we explore the relationship between the attack effectiveness in relation to number of data points used to compute the client updates. In this ablation study we consider the impact of number of instances a client owns, which allows the client to perform more (fewer) gradient steps before each communication round. We use CIFAR-100 dataset with AlexNet as an example. The number of clients is 10 and the total training set size is 40000. We divide this total training set into 10 different disjoint subsets of sizes: 400 instances, 1200 instances, up to 7600 instances. Each client gets one of these data subsets and will perform one local gradient step for each minibatch of 100 instances of private data. Therefore, the first client only performs 4 SGD steps and the last client performs 76 SGD steps. The evaluation of MI attacks against different clients (different number of gradient steps) is presented in Figure 7. In Figure 7(a), we can see that the AUC score decreases monotonically with the number of local gradient steps. In Figure 7(b), we can see that the TPR at $10^{-3}$ FPR is also decreasing when the number of steps increases. This is expected, since our attack is more effective when clients perform fewer gradient steps in local training and then send the parameter updates to the central server. If one client owns more private instances, it can perform more SGD (or Adam) steps, which reduces the efficacy of our attack.

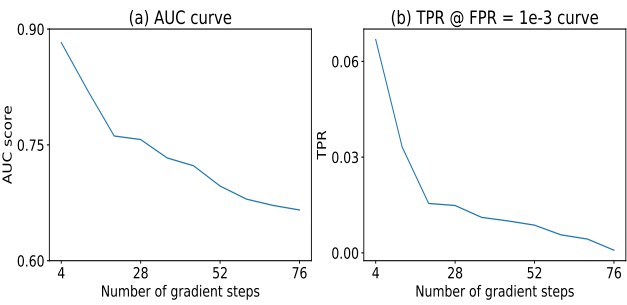

Figure 7: Ablation study on number of gradient steps per communication round. The cosine feature is used. CIFAR-100 dataset on AlexNet.

| Dataset-Model | Number of clients | Train Acc. | Test Acc. | Blackbox MI AUC |
|---|---|---|---|---|
| CIFAR-10, AlexNet | 1 | 0.9989 | 0.7346 | 0.7109 |
| CIFAR-10, AlexNet | 2 | 0.9968 | 0.7419 | 0.6898 |
| CIFAR-10, AlexNet | 4 | 0.9755 | 0.7446 | 0.6453 |
| CIFAR-10, AlexNet | 7 | 0.9062 | 0.7428 | 0.5989 |
| CIFAR-10, AlexNet | 10 | 0.8414 | 0.7450 | 0.5644 |
| CIFAR-10, AlexNet | 15 | 0.7771 | 0.7199 | 0.5362 |
| CIFAR-10, AlexNet | 20 | 0.7369 | 0.7038 | 0.5257 |
| CIFAR-100, AlexNet | 1 | 0.9939 | 0.3609 | 0.8791 |
| CIFAR-100, AlexNet | 2 | 0.9976 | 0.3746 | 0.8793 |
| CIFAR-100, AlexNet | 4 | 0.9952 | 0.3739 | 0.8695 |
| CIFAR-100, AlexNet | 7 | 0.9318 | 0.3599 | 0.8054 |
| CIFAR-100, AlexNet | 10 | 0.7262 | 0.3448 | 0.7270 |
| CIFAR-100, AlexNet | 15 | 0.4666 | 0.3234 | 0.5987 |
| CIFAR-100, AlexNet | 20 | 0.3810 | 0.2963 | 0.5633 |

Table 14: Experimental results of varying number of virtual clients with fixed data partitions. Results are averaged over 5 runs.

| Dataset-Model | Number of clients | Train Acc. | Test Acc. | Blackbox MI AUC |
|---|---|---|---|---|
| CIFAR-10, AlexNet | 1 | 0.9989 | 0.7346 | 0.7109 |
| CIFAR-10, AlexNet | 2 | 0.9973 | 0.7421 | 0.6928 |
| CIFAR-10, AlexNet | 4 | 0.9765 | 0.7439 | 0.6492 |
| CIFAR-10, AlexNet | 7 | 0.9020 | 0.7446 | 0.5963 |
| CIFAR-10, AlexNet | 10 | 0.8356 | 0.7446 | 0.5597 |
| CIFAR-10, AlexNet | 15 | 0.7761 | 0.7250 | 0.5394 |
| CIFAR-10, AlexNet | 20 | 0.7349 | 0.7016 | 0.5276 |
| CIFAR-100, AlexNet | 1 | 0.9939 | 0.3609 | 0.8791 |
| CIFAR-100, AlexNet | 2 | 0.9982 | 0.3762 | 0.8817 |
| CIFAR-100, AlexNet | 4 | 0.9969 | 0.3806 | 0.8669 |
| CIFAR-100, AlexNet | 7 | 0.9310 | 0.3657 | 0.8015 |
| CIFAR-100, AlexNet | 10 | 0.6768 | 0.3637 | 0.6885 |
| CIFAR-100, AlexNet | 15 | 0.4526 | 0.3340 | 0.5751 |
| CIFAR-100, AlexNet | 20 | 0.3714 | 0.3048 | 0.5437 |

Table 15: Experimental results of varying number of virtual clients with re-partitioning training data every epoch. Results are averaged over 5 runs.

| Dataset | Medical-MNIST | Pneumonia | Kidney | Covid | Skin | Retina | CIFAR-100 | CIFAR-100 | Purchase | Texas |
|---|---|---|---|---|---|---|---|---|---|---|
| Model | LeNet | AlexNet | AlexNet | AlexNet | AlexNet | AlexNet | AlexNet | DenseNet-BC(100,12) | 4-layer-FC-NN | 4-layer-FC-NN |
| Attacked Layer | Conv1 | Conv2 | Conv2 | Conv2 | Conv4 | Conv3 | Conv5 | Conv66 | FC1 | FC1 |

Table 16: Selected layers to perform our proposed attacks for each dataset-model combinations. Models are trained from scratch.

| Dataset | Skin | Retina | CIFAR-100 | CIFAR-100 | Purchase | Texas |
|---|---|---|---|---|---|---|
| Model | AlexNet | AlexNet | AlexNet | DenseNet-BC(100,12) | 4-layer-FC-NN | 4-layer-FC-NN |
| Grad-norm | 1e10 | 1e10 | 1e10 | 1e10 | 1e10 | 1e10 |
| Noise-level | 1e-10 | 2e-12 | 2.5e-10 | 2.5e-11 | 3e-12 | 1e-12 |

Table 17: Hyperparameters used for DP-SGD defense. Note that different from using a meaningful privacy budget (e.g. $5$ or $10$), we choose to use much larger privacy budget. Our the goal is to see if DP-SGD can achieve meaningful empirical defense without causing testing accuracy drop more than $1\%$. The grad-norm stands for the upper bound for gradient of one individual instance.

## A.6 Selected layers to perform our proposed attacks

In this subsection we list all the selected layers to perform our proposed attacks for all different dataset-model combinations. For models trained from scratch, all the layers are listed in Table 16. For model trained using transfer learning, the last fully-connected layer is used, since we only fine-tune the last fully-connected layer.

## A.7 Ablation study on data augmentation

We perform an ablation study on data augmentation to see the impact of different data augmentation method over the effectiveness of our proposed attacks. For random-crop, we set the padding parameter to be $4$ and the size parameter to be $32$ (cropping the padded image into $32 \times 32$). For horizontal flip and vertical flip, we set the probability to be $0.5$. The results are summarized in Table 18. Cosine attack remains effective under the common data augmentation methods listed in Table 18.

## A.8 Overparameterization ablation study on ResNet by varying layer width

In this ablation study, we study the relationship between overparameterization level and effectiveness of our proposed attack. We use ResNet18 as an example model and the original number of filters for all four basic blocks are [64,128,256,512]. We change the overparameterization level of ResNet18 by varying the width convolutional layer (i.e. changing the number of filters for each convolutional layer) and making sure that all convolutional layers have the same number of filters. The result is summarized in Figure 8. We conclude that the attack effectiveness increases as the level of overparameterization increases.

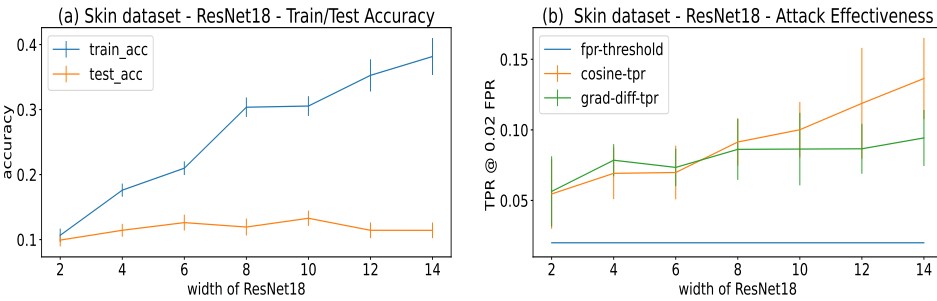

Figure 8: Ablation study on ResNet18 using different layer widths over Skin dataset. Averaged over 4 runs.

| Data augmentation method | centering cosine attack PLR (FPR @ 2%) ↑ | gradient-diff attack PLR (FPR @ 2%) ↑ |
|---|---|---|
| center-crop (no data augmentation defense) | 7.26±1.54 | 4.48±1.21 |
| random-crop | 7.35±1.63 | 3.90±0.54 |
| center-crop + horizontal flip | 8.50±1.93 | 1.80±0.79 |
| random-crop + horizontal flip | 9.55±3.22 | 5.74±1.83 |
| random-crop + horizontal flip + vertical flip | 6.39±1.91 | 5.91±1.16 |

Table 18: Ablation study on data augmentation. CIFAR-100 dataset with AlexNet.

## A.9  FURTHER RELATED WORKS

**Existing membership inference attacks in federated learning that need access to private members.** Nasr et al. (2019) presented the very first analysis on membership inference attack against neural networks under federated learning with access to some training members. The adversary can be either the central server or one of the participants in the federated learning framework. Besides, they presented the definition of passive attacker and active attacker, where the passive attacker does the training normally and the active attacker breaks the federated learning protocols to improve the effectiveness of MI attacks. For passive attackers, they showed that the attackers can take advantage of the gradients, activation maps, prediction vectors, loss and true label of one instance using local model of each user at different epochs during the training process and perform MI attacks. An attack model is trained using some known members. On the other hand, two active attacker strategies were proposed. The first one is called **gradient ascent attacker**, which means that the attacker will do gradient ascent on targeted samples. By taking the action of gradient ascent, the loss of the targeted samples will be increased. For member instances, their loss will be abruptly decreased by one client, thus can be distinguished from non-members. Both central server adversary and client adversary can utilize this strategy. The second active attack method is called **isolating attacker**. The attacker can isolate one special target participant by creating a separated copy of the aggregated model. This is equivalent to training a model on a single machine using the data from the targeted client. To use this isolation attack, the adversary needs to be the central server. Zari et al. (2021) proposed an efficient passive membership inference attack under federated learning which also requires some known members. Their attack uses as the feature vector the probabilities of the correct label under local models at different epochs, namely $\langle F_{\theta_{C_i}^{(1)}}(x)_y, F_{\theta_{C_i}^{(2)}}(x)_y, \cdots, F_{\theta_{C_i}^{(T)}}(x)_y \rangle$, where $T$ is the number of communication rounds. An attack model is trained to predict membership using known members and known non-members (which come from validation). Since the feature vector contains only one number from one epoch, this attack requires significantly less computational resources than the Nasr's attack which uses information such as gradient and activation maps. The authors showed that using this probability feature vector can achieve higher membership inference attack accuracy than Nasr's attack on CIFAR-100 dataset with AlexNet and DenseNet, but lower accuracy than Nasr's attack on Purchase dataset. Melis et al. (2019) identified membership leakage when using FL for training a word embedding function, which is a deterministic function that maps each word to a high-dimensional vector. Given a training batch (composed of sentences), the gradients are all 0's for the words that do not appear in this batch, and thus the set of words used in the training sentences can be inferred. The attack assumes that the participants update the central server after each mini-batch, as opposed to updating after each training epoch.

**Existing blackbox membership inference attacks in (non-federated) centralized training settings.** Shokri et al. (2017) presented the one of the earliest studies on MI attacks against classifiers. They proposed to use the prediction vector from the trained classifier as the attack feature to infer the membership of one given instance. Moreover, the attacker is assumed to know the model architecture, the model training hyperparameters and have access to a large dataset that comes from the same data distribution as the training data of the targeted classifier. With all these information, the attacker could train **shadow models** to simulate the training process and collect attack related information to train their attack models. Yeom et al. (2018) presented a theoretical analysis on privacy leakage based on prediction loss. They showed that the generalization gap between training accuracy and testing accuracy could serve as a lower bound for MI attack accuracy, as the attacker predicts member iff the model's prediction is correct. Besides, they proposed a threshold attack based on prediction loss, and the attacker predicts member iff the prediction loss is lower than the threshold, since the loss of members is generally lower than the loss of non-members. Jayaraman et al. (2021) suggested one MI attack based on Gaussian noise. The intuition is that for a member

instance, the prediction loss should increase after adding random noise. This MI attack adds multiple different random noise to the given instance and count how many times the prediction loss of the noisy instance is higher than the prediction loss of the original instance. The given instance is predicted to be a member if the count is beyond a threshold set by the attacker.

Inspired by Yeom et al. (2018), Song et al. (2019) proposed to adjust the prediction loss by a class-dependent value by noticing that some classes are harder to correctly classify than other classes. In addition, Song et al. (2019) proposed to use a modified prediction entropy to predict membership. The modified prediction entropy is calculated using this formula: $Mentr(F_\theta(x), y) = -(1 - F_\theta(x)_y) \log(F_\theta(x)_y) - \sum_{i \neq y} F_\theta(x)_i \log(1 - F_\theta(x)_i)$. Shadow models are needed to select the class-dependent loss adjustments. Sablayrolles et al. (2019) proposed to consider per-example hardness, which set a particular loss threshold for one instance. One instance is predicted to be a member if its loss is higher than this per-example loss threshold. Shadow models are needed to select this per-example loss threshold. It is also proven in Sablayrolles et al. (2019) that white-box MI (with access to model parameters) attack is no more effective than blackbox attack when the posterior distribution of model parameters (after training on a given training set) follows $\mathcal{P}(\theta | x_1, ... x_n) \propto e^{-\frac{1}{T} \sum_{i=1}^{n} \mathcal{L}(\theta, x_i)}$.

Watson et al. (2021) suggested to set the per-example loss threshold by the average of loss of one example on shadow models that are not trained using this example. Ye et al. (2021) followed the same shadow model procedure as Watson et al. (2021) to produce a collection of loss for one instance when this instance is not used in the training. Next, a one-sided hypothesis testing is proposed to predict membership. The advantage of this hypothesis testing is that the attacker could select a false positive rate. However, the possible false positive rate range is limited by the number of shadow models. Carlini et al. (2021) argued that the average-case metrics such as accuracy and AUC score fail to characterize whether the attack can confidently identify any members of the training set. To confidently identify members, true positive rate at a fixed low false positive rate (e.g. $0.1\%$) should be used to evaluate the effectiveness of different attacks. Carlini et al. (2021) also proposed a new parametric approach to explore the attack effectiveness at a very low false positive rate. This attack uses shadow models to collect two set of loss for each particular instance $x$ (one member set of loss when this instance is member and one non-member set of loss when this instance is a non-member). Next, the probability of the true label $F_\theta(x)_y$ is transformed by $\log(\frac{F_\theta(x)_y}{1 - F_\theta(x)_y})$, so the distribution of transformed probabilities of both sets look close to normal distribution. To predict membership of a given prediction $F_\theta(x)$, the attacker could calculate the two probabilities: the probability that $F_\theta(x)_y$ is drawn from the member set and the probability that $F_\theta(x)_y$ is drawn from the non-member set (by assuming the distributions of both sets are normal). The ratio of these two probabilities is then used as the metric to predict membership. Hui et al. (2021) exploits the model's prediction in a special "set" fashion. The adversary is aware of a set $M$ of members, and $N$ of non-members, and compute $d_1 = D(M \cup \{x\}, N)$ and $d_2 = D(M, N \cup \{x\})$, where $D$ denotes the *Maximum Mean Discrepancy (MMD)* (Fortet & Mourier, 1953) of two distributions of features extracted from the prediction vectors. The intuition is that if $x$ is a member, then we tend to have $d_1 > d_2$, and if $x$ is non-member, we tend to have $d_1 < d_2$. This attack only provides predicted membership, which means it is infeasible to evaluate its effectiveness at a low false positive rate.

In summary, existing blackbox MI attacks heavily rely on shadow models, which requires the attacker to have access to a large validation set (at least as large as the training set). However, for medical images, this is typically unrealistic. Therefore, in our baslines we focus on blackbox loss attacks from Yeom et al. (2018), the Gaussian noise attack called Merlin from Jayaraman et al. (2020) and the modified prediction entropy attack from Song et al. (2019).

**Defenses against blackbox MI attacks in centralized setting.** Distillation was proposed in Hinton et al. (2015) for the purpose of model compression, and used inShejwalkar & Houmansadr (2021) as a defense against MI attacks. InShejwalkar & Houmansadr (2021), one first trains a teacher model using the private training set, and then trains a student model using another unlabeled dataset from the same distribution as the private set. The training objective is to minimize the KL-divergence between the student model's output and the probability vector predicted by the teacher model under a pre-defined temperature $T$ in softmax function. The student model is given as the output. The intuition is that since the student model is not directly optimized over the private set, their membership may be protected. In Kaya et al. (2020), the authors compared the distillation technique with other regularization techniques. However, the most effective attacks were not considered in the

evaluation inKaya et al. (2020); Shejwalkar & Houmansadr (2021). Also, membership of the second (unlabelled) dataset is not considered.

Huang et al. (2021) proposed DAMIA to defend membership inference attack. DAMIA adopt the domain adaptation technique from transfer learning and directly apply it to the training process as described in Tzeng et al. (2014). DAMIA is able to reduce the attack accuracy of membership inference to be less than $60\%$. However, the success of DAMIA comes with significant testing accuracy drops.

Wang et al. (2021) proposed that weight pruning can be utilized to defend membership inference attack. By applying weight pruning, the training accuracy would be decreased while the testing accuracy is maintained. However, the compression rates of different layers are different, making the weight pruning technique hard to be generalized to new models. For models with a large amount of redundant parameters such as VGG, weight pruning can be helpful; however when it comes to compressed model such as MobileNet, weight pruning will bring significant reduction on testing accuracy.

Chang et al. (2019) proposed a defensive federated learning scheme in knowledge transfer fashion. After a few rounds of local training on their private data, the Cronus parties share their predictions on the public data. This new training strategy can defend model poisoning attack effectively. However, the availability of a large set of public data might hinder the deployment. Another drawback of this training scheme is the testing accuracy drop.

**Membership inference attacks in more settings.** Liu et al. (2021) evaluated the privacy leakage of pre-trained models which are trained using unsupervised contrastive learning strategy. The authors proposed a specific membership inference attack against pre-trained models. Given one instance, this new attack generates many perturbed versions of this instance and gather all the embeddings of these perturbed versions using the pre-trained model. The intuition is if one instance is used in the training of this pre-trained model, then the embeddings of its perturbed versions are generally closer to each other. He & Zhang (2021) utilized the same contrastive learning idea and proposed to fine-tune the pre-trained model to get the final target model. Experiments showed that using pre-trained model as feature extractor would reduce privacy leakage, comparing to training models from scratch.

Hidano et al. (2020) evaluated MI attack under the setting where the attacker can know the parameters of some shallow layers. In the experiment, the author assumed that the attacker can get all but the last layer and the parameters of these layers are used to initialize shadow models to facilitate the class-vector attack. With these known parameters, the class-vector attack can outperform its black-box version.

**Other types of privacy leakage in deep neural networks.** Fredrikson et al. (2015) studied the model inversion attack based on the model predictions. Model inversion attacks recover aggregate details of particular sub-classes instead of individual training examples.

Hitaj et al. (2017) proposed a model inversion attack to reconstruct class representative. This attack uses a GAN to generate class representative and the discriminator is initialized using the model downloaded from the server.

Wei et al. (2020) presented a instance-reconstruction attack under federated learning setting. The authors assumed that the batch size is 1 when the clients do their local training. Given the gradients of one instance, the attacker can reconstruct the target instance by gradient descent. The attacker starts from a blank image and iteratively update the blank image so that the gradients of this image can match up with the given gradients. If the batch size grows up, the proposed attack is going to reconstruct class representatives.

Salem et al. (2020) studied the possible information leakage of an update set when the machine learning model is updated using the update set. To detect the information leakage, the authors proposed two different attacks: label inference attack and instance reconstruction attack. These two attacks can be applied to both single instance update set case and multi-instances update set case, with only black-box access to the machine learning model.

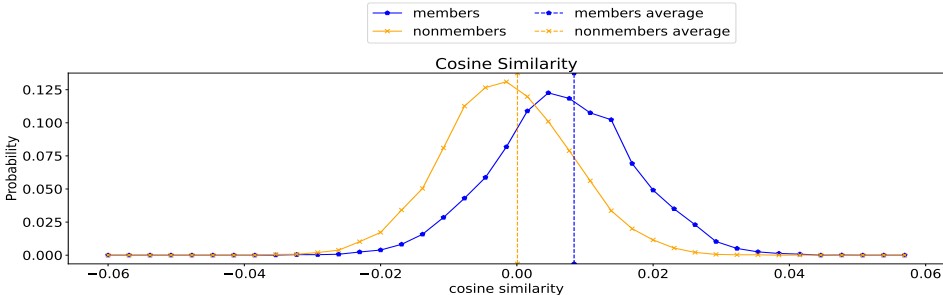

Figure 9: Cosine similarity distribution for members and non-members, averaged over the last 200 epochs. CIFAR-100 dataset, AlexNet. The average of members cosine similarity is 0.00839, the average of nonmembers cosine similarity is 0.00006. The standard deviation of members cosine similarity is 0.01028; for nonmembers, the standard deviation is 0.00938.

Zhao et al. (2020) proposed one label estimation method based on the gradients of one instance. This label estimation method could provide perfect prediction. With improved label estimation, the reconstruction attack from Zhu & Han (2020) can reach higher fidelity.

Zhu & Han (2020) presented a gradient-based instance reconstruction attack. If the gradients of one specific instance are revealed to one adversary who can access the trained model, then the adversary is able to reconstruct the specific instance with high fidelity. One random instance is gradually optimized by matching its gradients with the provided gradients of the specific instance. For gradients of one batch of instances, when the batch size is small, this reconstruction attack can be extended to sequentially optimize different random inputs, so that the average gradients will match the gradients of this batch.

Geiping et al. (2020) noticed that, when the attacker tries to reconstruct the input by gradient matching, the direction of gradients is more important than the norm of the gradients. Thus, the authors proposed to use a cosine similarity based cost function to optimize a random input so that the direction of this random input is the same as the gradients of the target instance which one tries to reconstruct.

Yin et al. (2021) extended previous gradient-based instance reconstruction attack to ImageNet. By using the label estimation method from Zhao et al. (2020), they proposed a label estimation method for batch of images when the batch size is smaller than the number of classes and every class only appears once in the batch. After getting the accurate labels, they followed the image reconstruction method from Yin et al. (2020) and a few regularizations are used to improve the quality of the reconstructed images.

Maini et al. (2021) brought a new research question: dataset inference. Dataset inference stands for the process of identifying whether a suspected model copy has private knowledge from the original model's dataset, as a defense against model stealing. Since the defender is granted with access to the original training set, the defender can launch membership inference attack for every instance in the training set against the suspected model. As long as the membership inference attack could achieve somewhat better performance than random guessing, notice that the size of the training set is usually huge, the defender still stands a great chance to launch dataset inference successfully. The authors proposed to use the distance to the classification boundary as the feature of each instance to infer membership.

Haim et al. (2022) proposes an interesting set of data reconstruction attacks based on gradient decomposition of trained models. Training data reconstruction can be used as a membership inference attack. Our attack can be seen as belonging to a general family of gradient-based attacks of overparameterized models, albeit with a distinct methodology from Haim et al. (2022) (we take advantage of the orthogonality of random vectors to make a fast and simple attack).

| | Skin AlexNet | Retina AlexNet | CIFAR-100 AlexNet | CIFAR-100 DenseNet | |
|---|---|---|---|---|---|
| | PLR ↑ (FPR @ 2%) | PLR ↑ (FPR @ 1%) | PLR ↑ (FPR @ 1%) | PLR ↑ (FPR @ 1%) | fr |
| blackbox loss (Yeom et al., 2018) | 1.02±0.29 | 0.50±0.03 | 1.13±0.23 | 0.70±0.14 | |
| **fed-loss attack** | 1.45±0.37 | 1.02±0.37 | 1.01±0.04 | 0.70±0.13 | |
| **cosine attack** | 3.50±1.30 | 1.85±0.30 | 4.20±0.84 | 3.60±1.03 | |
| **gradient-diff attack** | 5.50±1.08 | 0.50±0.08 | 0.00±0.00 | 0.00±0.00 | |

Table 19: Evaluation of our cosine attack, gradient-diff attack and the blackbox loss attack without private information on hard medical image datasets and benchmark image dataset CIFAR-100. Results are averaged over 10 participants. The attacker can only observe the communication from the central server to clients (i.e. only the aggregated model is available to the attacker).

| | Pneumonia AlexNet | Kidney AlexNet | Covid AlexNet |
|---|---|---|---|
| | PLR ↑ (FPR @ 5%) | PLR ↑ (FPR @ 2.5%) | PLR ↑ (FPR @ 1%) |
| blackbox loss (Yeom et al., 2018) | 0.25±0.39 | 0.58±0.43 | 0.40±0.43 |
| **fed-loss attack** | 0.96±0.52 | 1.10±0.32 | 0.64±0.38 |
| **cosine attack** | 1.20±0.37 | 4.02±1.05 | 2.25±1.08 |
| **gradient-diff attack** | 0.60±0.48 | 0.93±0.21 | 0.60±0.28 |

Table 20: Evaluation of our cosine attack, gradient-diff attack and the blackbox loss attack without private information on easy medical image datasets. Results are averaged over 10 participants. The attacker can only observe the communication from the central server to clients (i.e. only the aggregated model is available to the attacker).

