# OpenReview forum: "Effective passive membership inference attacks in federated learning against overparameterized models"
_ICLR.cc/2023/Conference — ICLR 2023 poster_

### Official Review · Reviewer_nM4b · 2022-10-18

**Confidence:** 4
**Clarity, Quality, Novelty And Reproducibility:** good
**Correctness:** 3
**Technical Novelty And Significance:** 3
**Empirical Novelty And Significance:** 3
**Recommendation:** 6

**Strength And Weaknesses:**

Strengths:

- Trendy topic
- New perspective on membership inference
- Easy to follow

Weaknesses:

- Insufficient explanations for the intuition
- Unbalanced validation dataset for the MIA

Comments for the authors:

This paper proposes a new passive membership inference attack in federated learning against overparameterized models. The intuition for the proposed attack method is novel and it is easy to implement such attacks. The experimental results also show the effectiveness of this method.

However, I have the following concerns:

- The intuition needs more explanations. Figure 1 shows the distributions of pair-wise cosine similarity of gradients from different instances. However, it seems that there is no big difference between the distributions of members and non-members. So why the cosine similarity can serve as a feature to discriminate members from non-members? I would suggest the authors add more explanations here.

- The validation dataset is biased. The validation dataset used by the attack model only contains non-member instances. Is there any method to prevent the attack model from giving biased predictions? For example, if the attack model always predicts the data samples as non-members, this attack model will perform perfectly on the validation dataset, which does not mean that this attack model can give correct predictions to members as well. I suggest the authors give some explanations here and improve the original method if possible.

- Will the data augmentation strategies (e.g., RandomCrop) in the clients affect the attack effectiveness? I am curious about whether some data augmentation strategies will affect the similarity between the client updates and the gradients of data samples. I would suggest the authors also complement the experiments for this ablation study.

- This paper claims that the proposed method will not work well in small neural network models in Section 1. However, in Section 4, the authors also evaluate the relationship between the model overparameterization level and attack effectiveness and it seems that the attack is still effective (though with lower performance) even when the model has relatively low model overparameterization levels. Then why this method does not work well only for small models? I would suggest the authors give more analysis for it and complement the corresponding ablation study if necessary.

Minor:

- On Page 2, "iff $u \in \mathcal{O}^\prime$" -> "if $u \in \mathcal{O}^\prime$"


**Summary Of The Paper:**

This paper proposes a new passive membership inference attack in federated learning against overparameterized models. The attacker has no access to the private training data information and any other data instances (e.g., shadow dataset) sampled from the same data distribution. The proposed attack method is based on the similarity between client updates and gradients of instances. The evaluations demonstrate the effectiveness of this method.


**Summary Of The Review:**

see strength and weakness

---

> ### Author Response · Authors · 2022-11-18
> **Response to Reviewer nM4b**
>
> We thank the reviewer for the support and valuable feedback. We have updated our draft based on your feedback. Our answers are below:
>
> **Q1:** The intuition needs more explanations. Figure 1 shows the distributions of pair-wise cosine similarity of gradients from different instances. However, it seems that there is no big difference between the distributions of members and non-members. So why the cosine similarity can serve as a feature to discriminate members from non-members? I would suggest the authors add more explanations here.
>
> **A1:** Thanks, we emphasized the following in the draft. Figure 1(a) shows the cosine similarity (cosim) between gradients of two *distinct* member instances, Figure 1(b) a non-member and a member, and Figure 1(c) two *distinct* non-member instances. This is to show that gradients of distinct instances are approximately orthogonal no matter whether they are members or non-members.
>
> **Q2:** The validation dataset is biased. The validation dataset used by the attack model only contains non-member instances. Is there any method to prevent the attack model from giving biased predictions? For example, if the attack model always predicts the data samples as non-members, this attack model will perform perfectly on the validation dataset, which does not mean that this attack model can give correct predictions to members as well. I suggest the authors give some explanations here and improve the original method if possible.
>
> **A2:** There is no bias. In the updated draft we have clarified that the only part of our attack that needs the non-member data is the calibration of $\gamma$ in Eq (4) using a conformal hypothesis test. In the hypothesis test we just need to set $\gamma$ as the rejection region of the null model "non-member". And declare an instance as a member if it is rejected as non-member.
>
> **Q3:** Will the data augmentation strategies (e.g., RandomCrop) in the clients affect the attack effectiveness? I am curious about whether some data augmentation strategies will affect the similarity between the client updates and the gradients of data samples. I would suggest the authors also complement the experiments for this ablation study.
>
> **A3:** That is a great suggestion. Thank you. In the updated draft we have added new results in Section A.7 in the Appendix showing that the cosine attack remains effective under the common data augmentation methods (e.g., RandomCrop) listed in Table 18.
>
> **Q4:** This paper claims that the proposed method will not work well in small neural network models in Section 1. However, in Section 4, the authors also evaluate the relationship between the model overparameterization level and attack effectiveness and it seems that the attack is still effective (though with lower performance) even when the model has relatively low model overparameterization levels. Then why this method does not work well only for small models? I would suggest the authors give more analysis for it and complement the corresponding ablation study if necessary.
>
> **A4:** Thank you for your suggestions. The effectiveness of the attack increases as the model becomes more overparameterized. We have added a new ablation study in Section A.8 in the Appendix, with results shown in Figure 8 (besides the ablation study in Figure 3(b) which used matrix rank, and gives similar results). The orthogonality of independent isotropic random vectors happens only in high dimensions, hence, we had conjectured that the effectiveness should be correlated with the dimension of the parameters. And indeed, in both ablation studies we see this.

---

### Official Review · Reviewer_o3B4 · 2022-10-24

**Confidence:** 4
**Correctness:** 2
**Technical Novelty And Significance:** 3
**Empirical Novelty And Significance:** 3
**Recommendation:** 6

**Clarity, Quality, Novelty And Reproducibility:**

The clarity and quality of writing needs to be improved. The use of gradient orthogonality for membership inference is novel. The detailed hyperparameters were described in the appendix but no code or pseudocode of the attacks were provided.


**Strength And Weaknesses:**

Strengths

1. The main observation about gradient orthogonality and the use of it in membership inference is novel.
2. Experiments covered a different range of datasets and defense methods.

Weaknesses

1. This paper is not very organized and some parts are confusing to read. For example, equation (2) is hard to parse due to the superscript, and half-step update is not clearly defined.
2. The threat model is particularly not clear. What is the capability of the attacker? What exactly does the attacker observe, the individual gradients for each client or the summed gradients overall clients? How do you motivate such a threat model?
3. The implementation details of differential privacy were missing. It’s very surprising that Table 5 showed the ineffectiveness of DP as it is contradicting the DP theoretical guarantee. What are the privacy budgets? Did you perform the attack on the individual gradients with local DP or did you perform the attack on the aggregated gradients with central DP? Also, training with DP on datasets like CIFAR100 with less than 1% drop in utility is very hard, and the authors should provide more details to support their claims that such a defense strategy is meaningful.

Minor: Section 2.1, federated averaging after equation 1 should be an average but currently it is a sum.


**Summary Of The Paper:**

This paper proposes membership inference attacks in federated learning for overparameterized models. The main observation is that in later stages of training an overparameterized model, the gradients of different examples will become orthogonal and thus from the dot product between gradients, the attacker can infer the membership. The attacks were evaluated on multiple datasets and different mitigation strategies were discussed and implemented.


**Summary Of The Review:**

I don’t think the paper is ready for publication given the weaknesses discussed above.

---

> ### Author Response · Authors · 2022-11-18
> **Response to Reviewer o3B4**
>
> Thanks for the suggestions and comments. If the reviewer wants further clarifications on other parts, please let us know, we will happily do that. Please see our answers below:
>
> **Q1:** This paper is not very organized and some parts are confusing to read. For example, equation (2) is hard to parse due to the superscript, and half-step update is not clearly defined.
>
> **A1:** The half-steps are the mini-batches used to update the local model. We have clarified this in the updated draft (in blue). If the reviewer wants further clarifications on other parts, please let us know, we will happily do that.
>
> **Q2:** The threat model is particularly not clear. What is the capability of the attacker? What exactly does the attacker observe, the individual gradients for each client or the summed gradients overall clients? How do you motivate such a threat model?
>
> **A2:** Thanks for the question. We had the threat model described throughout the paper but it is a good point to consolidate it at a single place. We have done so in the updated draft (see in blue). It is reproduced below for your convenience.
>
> **Threat model:** We consider a passive eavesdropping attacker, who is able to observe updated model parameters from rounds of communication between server and clients. The attacker may observe either (a) the server → client (global) model update, (b) client → server model updates, or (c) both. In our experiments we assume eavesdropping of client → server model updates. Different from most existing works which assume access to private member data instances (Nasr et al., 2019; Zari et al., 2021), we assume that the attacker has access to a small validation dataset which contains only non-member data instances. Our work is motivated by the strict data handling protections of medical records, creating attack vectors that do not rely on having access to private medical data.
>
> **Q3:** "The implementation details of differential privacy were missing. It’s very surprising that Table 5 showed the ineffectiveness of DP as it is contradicting the DP theoretical guarantee. What are the privacy budgets? Did you perform the attack on the individual gradients with local DP or did you perform the attack on the aggregated gradients with central DP? Also, training with DP on datasets like CIFAR100 with less than 1% drop in utility is very hard, and the authors should provide more details to support their claims that such a defense strategy is meaningful."
>
> **A3:** The full details are in Section 2.5 and Section 4, which we reproduce at the end of this comment for completeness. The DP parameters are given in Table 17 in the updated draft in the Appendix. Our the goal is to see if DP-SGD could achieve meaningful empirical defense without causing testing accuracy drop more than 1%. The DP guarantees of (Abadi et al., 2016) are on the final classifier, not on an attack on the gradients. In the optimization itself see "Algorithm 1 Differentially private SGD (Outline)" in (Abadi et al., 2016) adds random Gaussian noise to the gradient, which we know is not an effective defense against our attack due to independent isotropic random vector orthogonality.
>
> A more in-depth description can be found in the updated draft but here is a snippet for your convenience: *DP-SGD (Abadi et al., 2016) adds noise to the gradients (or other components) during the training process. Training with differential privacy provides theoretical and worst-case guarantee against any MI attacks over the final classifier. Achieving a meaningful theoretical guarantee (i.e., with a reasonably small ε, e.g., < 5) requires the usage of very large noises, resulting significant testing accuracy drop, and provide limited defense against our gradient attack (due to the independent isotropic random vector properties of the Gaussian noise used by (Abadi et al., 2016)). *
>
> **Q4:** Section 2.1, federated averaging after equation 1 should be an average but currently it is a sum.
> **A4:** Thanks! It was a typo. Fixed in the updated draft.

---

> > ### Author Response · Authors · 2022-11-18
> > **Response to Reviewer o3B4**
> >
> > **Q5:** "no code or pseudocode of the attacks were provided""
> >
> > **A5:** The algorithm of our attack is extremely simple. We rewrote Eq (4) to make it clearer in the updated draft. The cosine similarity attack is just applying Eq (4) (we choose $\gamma$ based on our conformal hypothesis test). If we can eavesdrop over multiple communication rounds, we just need to average the cossim over all rounds (as described later). The choice of $\gamma$ is also simple. We have expanded the text to consolidate the description of the conformal hypothesis test we use to obtain a calibrated false positive rate (in blue). It also explains why we do not advise using a Student's t-test for membership inference (because of the inability to define a false positive rate). If the reviewer thinks the conformal hypothesis test is not clear, we can write a simple algorithm in the rebuttal and camera-ready.

---

### Official Review · Reviewer_61U3 · 2022-10-24

**Confidence:** 4
**Correctness:** 4
**Technical Novelty And Significance:** 3
**Empirical Novelty And Significance:** 4
**Recommendation:** 8

**Clarity, Quality, Novelty And Reproducibility:**

I found the writing in this work to be clear. While the related work section in the main body is short, the authors do supplement it with a larger overview in the appendix for interested readers.

Concerning additional related work, the authors might find Haim et al, "Reconstructing Training Data from Trained Neural Networks" interesting, which describes a series of data reconstruction attacks based on gradient decompositions of trained models. Overall, for me, the connection of this attack to data reconstruction attacks is particularly interesting, for example, the proposed attack can be seen as a zero-order evaluation of the gradient inversion objectives based on cosine similarity, evaluating on the data point to be tested.

**Strength And Weaknesses:**

I'm a fan of this work. The proposed approach is straight-forward and well-motivated and after reading this submission I am convinced that this is a strong baseline membership inference attack for federated learning. I have a few minor questions concerning ablations, notes about related defenses and minor remarks which I will list below, but overall my impression is positive.

* One defense against this attack thatis not well evaluated in the main body is aggregation. I think it would be great to consider aggregation (i.e. the server update is aggregated over more data, for example from multiple users through secure aggregation) as a central defense in Section 4, as this would be to the most likely encountered defense mechanism in practice. I see some discussion in Table 8 which I think is related, but it would be great to include a plot of the effectiveness of the attack in relation to the number of data points used to compute the attacked update.

* Concerning the validation of overparametrization: While the rank experiment in Fig. 3 is interesting, and certainly a valid case of overparametrization, to me, other experiments would be closer to the convential understanding of overparametrization: One example could be to use a ResNet model on CIFAR-100, but to scale the width of all layers equally.

* The one thing I don't understand about Section 2.3 is the summation over $a \in \mathbb{Y}$. For a conventional membership inference attack I would assume the data point to be tested to be a pair $(x,a)$ containg both image and label. Here, the label seems to be an additional unknown to the attacker? It would be great to briefly formalize the threat model some where in Section 2 and clarify this.

* Concerning ablations, the attack is introduced targeting only one layer in the model. This is ablated in Fig.2 (for what seems to be AlexNet), but I couldn't find information which layer is targeted in other settings. I see the automated choice based on non-member information, I am basically just interested in the outcome. Also, a reasonable extension of Fig.2b) for me would include the attack, but using the entire gradient, just as baseline.

**Summary Of The Paper:**

The submission titled "Effective passive membership inference attacks in federated learning against overparameterized models" describes a well-motivated black-box membership inference attack that can be instantiated in a federated learning setting, i.e. this is an inference from user update/gradient to dataset membership likelihood. The submission motivates the effectiveness of gradient alignment as a metric of membership inference success for large, overparametrized models and shows empirical success in a variety of simulated FL scenarios.

**Summary Of The Review:**

In summary, I believe this to be a significant contribution and recommend acceptance.

---

> ### Author Response · Authors · 2022-11-18
> **Response to Reviewer 61U3**
>
> Thank you for the excellent comments, reference, and support.
>
> **Q1** "One defense against this attack that is not well evaluated in the main body is aggregation. I think it would be great to consider aggregation (i.e. the server update is aggregated over more data, for example from multiple users through secure aggregation) as a central defense in Section 4, as this would be to the most likely encountered defense mechanism in practice. I see some discussion in Table 8 which I think is related, but it would be great to include a plot of the effectiveness of the attack in relation to the number of data points used to compute the attacked update."
>
> **A1:** That is a great question! This defense seems related to only being able to monitor server -> client model updates. We are re-running our experiments with this setting (these should be ready for the camera-ready). However, we already tested a harder scenario in Section A.5. We divided the training set into 10 different disjoint subsets of sizes: 400 instances, 1200 instances, up to 7600 instances. We then attacked the main model update as usual. Each client gets one of these data subsets and will perform one local gradient step for each minibatch of 100 instances of private data. Therefore, the first client only performs 4 SGD steps (less data) and the last client performs 76 SGD steps (more data). In Figure 7(a), we can see that the AUC score decreases monotonically with the number of local gradient steps. In Figure 7(b), we can see that the TPR at 10−3 FPR is also decreasing when the number of steps increases. This is expected, since our attack is more effective when clients perform fewer gradient steps in local training. If one client owns more private instances, it can perform more SGD (or Adam) steps, which reduces the efficacy of our attack as shown in the figure. While this is an effective defense, there is the drawback of training with loosely synchronized clients, that can perform 76 gradient steps independently, may not result in an accurate final model.
>
>
> **Q2:** "Concerning the validation of overparametrization: While the rank experiment in Fig. 3 is interesting, and certainly a valid case of overparametrization, to me, other experiments would be closer to the convential understanding of overparametrization: One example could be to use a ResNet model on CIFAR-100, but to scale the width of all layers equally."
>
> **A2:** Thank you for the suggestion!  We have added a **new** ablation study in Section A.8 in the Appendix, with results shown in Figure 8 (note that the results are similar to the matrix rank results in Figure 3(b)). Since independent isotropic random vectors become increasingly more orthogonal in high dimensions, we also expect that the effectiveness should be correlated with the dimension of the parameters (which it does in both ablation studies).
>
> **Q3:** "The one thing I don't understand about Section 2.3 is the summation over a."
>
> **A3:** We assume we may not have the same label for the instance as in the private data. If we do, the attack could be simplified to that specific label. We have added a sentence to clarify this in the draft.
>
> **Q4:** "Concerning ablations, the attack is introduced targeting only one layer in the model. This is ablated in Fig.2 (for what seems to be AlexNet), but I couldn't find information which layer is targeted in other settings. I see the automated choice based on non-member information, I am basically just interested in the outcome. Also, a reasonable extension of Fig.2b) for me would include the attack, but using the entire gradient, just as baseline.""
>
> **A4:** Thank you, that is a great suggestion. We have updated Figure 2 to include these cases. Using all parameters is clearly more effective but not that much more than the last CONV layer (using all parameters is significantly more computationally expensive).
>
> **Q5:** Add Haim et al, "Reconstructing Training Data from Trained Neural Networks" to related works.
> **A5:** Thanks for the Haim et al, 2022 reference! We added it to the other related work in the Appendix. The relationship between our work and their work is quite interesting and worth further exploration. For instance, how is gradient near-orthogonality tied to the desire to have more equations than variables in their method (overparametrization)? Can gradient near-orthogonality be used to reduce the computational burden of their approach? We can see a few avenues to explore.

---

> > ### Comment · Reviewer_61U3 · 2022-11-18
> > **Response**
> >
> > Thank you for your response. I appreciate the additional experiments and clarifications and have no further questions or concerns. I have read the other reviews and continue to think that this a good submission to ICLR.

---

> > > ### Author Response · Authors · 2022-11-20
> > > **Thank you**
> > >
> > > Thank you for your support! It means a lot to us since we deeply believe our work is important to the community.
> > >
> > > For instance, it is fascinating to think about the relationship between gradient orthogonality (our observations), lottery ticket hypothesis (early model sparsity in overparameterized models), and MI attacks in federated learning.
> > >
> > > 1. Federated learning seems to be key to the widespread adoption of ML models in healthcare and manufacturing, where no single entity has all the data but local data must remain private.
> > >
> > > 2. Model overparameterization seems to be key to the performance benefits of neural networks.
> > >
> > > We show 1 and 2 are seemly incompatible in the current state of affairs in federated learning (under our threat model). Can 1 and 2 provably coexist?
> > >
> > > Over time maybe our attack could be further refined (we are not sure how yet, but maybe something between our work, lottery tickets, and Haim et al, 2022?). Maybe to provably defend against our type of federated learning attack requires understand why overparameterization is key to the performance of neural networks. We think it is important to have our work published to encourage others to explore these topics.
> > >
> > > PS: We expect to have all the extra experiments related to your question Q1 done before the discussion phase ends. We are genuinely interested in knowing the answer. If we were to guess, it will have only moderate impact. E.g., 40 SGD steps per client (Fig 7) should be a much better defense than 10x the amount of data in the gradient sum for each client performing 4 SGD steps, and 40 SGD steps was not enough to eliminate the MI risk (it reduces effectiveness but the attack is still effective).

---

> > > > ### Author Response · Authors · 2022-11-30
> > > > **Results attacking global model**
> > > >
> > > > As promised, here are the experimental results when the attacker can only access to the global model ( the communication from central server to clients). For hard medical datasets (Skin, Retina) and CIFAR-100 dataset, the cosine attack also outperforms all other attacks, even though the overall effectiveness is reduced as expected due to less information available to the attacker. For easy medical datasets (Pneumonia, COVID, Kidney), we can see that the cosine attack is still effective as affective as before (attacking just the client's model).
> > > >
> > > >
> > > > |                                 |  Skin  |  Retina   |   CIFAR-100   | CIFAR-100|
> > > > | :------------------------------:|:-----------:|:---------:|:---------:|:---------:|
> > > > |                                 |   AlexNet   |  AlexNet  |  AlexNet  | DenseNet |
> > > > |                                 |     PLR     |    PLR    |   PLR     | PLR |
> > > > |                                 |  (FPR @ 2%)  | (FPR @ 1%) | (FPR @ 1%) |  (FPR @ 1%) |
> > > > | :------------------------------:|:-----------:|:---------:|:---------:|:---------:|
> > > > | Blackbox loss (Yoem et al. 2018)|  1.02  $\pm$  0.29  | 0.50 $\pm$  0.03 | 1.13 $\pm$  0.23 | 0.70 $\pm$  0.14 |
> > > > | fed-loss attack                 |  1.45  $\pm$ 0.37 | 1.02 $\pm$  0.37 | 1.01 $\pm$  0.04  | 0.70 $\pm$  0.13 |
> > > > | cosine attack                   |  3.50 $\pm$  1.30 | 1.85 $\pm$  0.30 | 4.20 $\pm$  0.84 | 3.60 $\pm$  1.03 |
> > > > | gradient-diff attack            | 5.50 $\pm$  1.08 | 0.50 $\pm$  0.08 | 0.00 $\pm$  0.00 | 0.00 $\pm$  0.00 |
> > > >
> > > >
> > > >
> > > > |                                 |  Pneumonia  |  Kidney   |   Covid   |
> > > > | :------------------------------:|:-----------:|:---------:|:---------:|
> > > > |                                 |   AlexNet   |  AlexNet  |  AlexNet  |
> > > > |                                 |     PLR     |    PLR    |   PLR     |
> > > > |                                 |  (FPR @ 5%) | (FPR @ 2.5%) | (FPR @ 1%) |
> > > > | :------------------------------:|:-----------:|:---------:|:---------:|
> > > > | Blackbox loss (Yoem et al. 2018)|  0.25  $\pm$  0.39  | 0.58  $\pm$  0.43 | 0.40  $\pm$ 0.43 |
> > > > | fed-loss attack                 |  0.96  $\pm$ 0.52 | 1.10  $\pm$ 0.32 | 0.64 $\pm$  0.38 |
> > > > | cosine attack                   |  1.20  $\pm$  0.37 | 4.02  $\pm$ 1.05 | 2.25  $\pm$ 1.08 |
> > > > | gradient-diff attack            |  0.60  $\pm$ 0.48 | 0.93  $\pm$ 0.21 | 0.60 $\pm$  0.28 |

---

### Decision · Program_Chairs · 2023-01-20

**Decision:**

Accept: poster

**Justification For Why Not Higher Score:**

Taking into account the strengths and weaknesses above and the reviewer comments, a poster decision seems adequate.

**Justification For Why Not Lower Score:**

Sufficient novelty to accept.

**Metareview: Summary, Strengths And Weaknesses:**

(a) The paper addresses the problem of membership inference attacks: namely, understanding whether a given data point is in the data set. Its key observation is that in over-parameterized models that are well trained, at some points gradient will become roughly orthogonal. And as a result, it will be possible to perform an attack by cosine similarity with the gradient of the queried
member.
(b) The interesting aspect of this paper is the link between over-parameterization and membership attacks, due to a certain orthogonal structure in the model. This seems new, and would be of interest to the federated learning and secure-learning audience. Experiments also support the results.
(c) The attack model could be made more realistic, although the reviewers thought it was acceptable for demonstrating the observations in the paper.

**Note From Pc:**

if the above contains the word "oral" or "spotlight" please see: "oral" presentation means -> notable-top-5% and "spotlight" means -> notable-top-25%. As stated in our emails, we are disassociating presentation type from AC recommendations

**Summary Of Ac-Reviewer Meeting:**

NA